# Xenosiderophore transporter gene expression and clade-specific filamentation in *Candida auris* killifish (*Aphanius dispar*) infection

Hugh Gifford [1], Tina Bedekovic[1], Nicolas Helmstetter[1], Jack Gregory [1], Qinxi Ma[1], Alexandra C. Brand [1], Duncan Wilson [1], Johanna Rhodes [2,3], Mark Ramsdale[1], Tetsuhiro Kudoh[4,5] & Rhys A. Farrer [1,5] ✉

*Candida auris* is a critical priority fungal pathogen (World Health Organization). Clinical management is challenging, with high associated mortality, rapidly increasing antifungal resistance, and frequent nosocomial outbreaks. A critical bottleneck in understanding virulence is the lack of gene expression profiling models during infection. We developed a fish embryo yolk-sac microinjection model using *Aphanius dispar* (Arabian killifish; AK) at human body temperature. This enabled interrogation of infection dynamics *via* dual host-pathogen RNA-seq across five major clades of *C. auris* (I-V). Host responses included heat shock, complement activation, and nutritional immunity, notably haem oxygenase (*HMOX*) expression during clade IV infection. We identified a pathogen transcriptional signature across all five clades, strongly enriched for putative xenosiderophore transmembrane transporter candidate (*XTC*) genes. We describe this family and a sub-clade of five putative haem transport-related (*HTR*) genes. Only the basal clade V isolate formed filaments, associated with canonical and atypical regulators of morphogenesis. Clades I and IV demonstrated increased virulence, accompanied by up-regulation of three *HTR* genes in clade IV, and the non-mating mating-type locus (*MTL*) gene *PIKA* in both clades. Our study provides insights into *C. auris* pathogenesis, highlighting species-wide in vivo up-regulation of *XTC* genes during host tissue infection.

*Candida auris* is an ascomycetous yeast that has rapidly progressed from a recently identified species to a World Health Organization (WHO) critical priority human fungal pathogen and global public health threat[1,2]. The earliest known isolate was retrospectively identified from a bloodstream infection dating to 1996[3], following its initial description as a novel species in 2008[4]. Over the past decade, *C. auris* has caused nosocomial outbreaks globally, leading to intensive care unit (ICU) closures and substantial infection control challenges and costs[5–7]. Alarmingly, *C. auris* is now a leading cause of invasive candidiasis in several hospital settings globally[8,9]. *C. auris* bloodstream infections carry an associated mortality rate of approximately 45%[10] and are often resistant to azole antifungals, with an increasing number of pan-drug resistant isolates reported globally[11]. Phylogenetic analysis has revealed five major clades of *C. auris*, and a minor sixth clade

recently found[12,13], suggesting multiple populations of *C. auris* have simultaneously emerged from unknown environmental reservoirs that have facilitated its rapid adaptation to human infection, drug resistance, and healthcare-associated persistence[14,15]. Clade-specific differences have been noted: clades I, III, and IV are more commonly associated with invasive disease, whereas clades II and V have been linked to otomycosis[16]. Mortality from candidaemia may be higher with clades I and IV than clade III, but this difference has not been shown to be statistically significant[17]. These differences in clinical presentation and virulence may reflect underlying genotypic variation across clades.

Gene expression profiling has become a standard approach for assessing host and pathogen responses to infection. However, no experiment has described the gene expression of *C. auris* during in vivo infection of living

[1]MRC Center for Medical Mycology, University of Exeter, Exeter, UK. [2]Department of Medical Microbiology, Radboudumc, Nijmegen, The Netherlands. [3]MRC Centre for Global Infectious Disease Analysis, Imperial College London South Kensington Campus, London, UK. [4]Biosciences, University of Exeter, Exeter, UK. [5]These authors jointly supervised this work: Tetsuhiro Kudoh, Rhys A. Farrer ✉e-mail: R.Farrer@exeter.ac.uk

tissue to date. Studies have assessed the pathogen transcriptome in an in vivo murine catheter model[18], human whole blood ex vivo[19], co-incubation with human dermal fibroblasts[20] or murine cell line-derived macrophages[21]. Additionally, host responses have been assessed in murine peripheral skin[22], human peripheral blood-derived monocytes[23], and murine bone marrow-derived macrophages[24]. *C. auris* has been shown to reproduce as budding yeasts in mammalian tissues[25,26] and possess multiple adhesins associated with an aggregation phenotype[27,28]. There has been no evidence of toxin synthesis, unlike distantly related *C. albicans*[29,30]. *C. auris* digests of host tissue through secreted aspartyl proteases[31] and evades host immune responses through cell surface masking[32] and intra-phagocytic survival with cell lysis[33]. However, these studies have not interrogated in vivo in-host pathogen gene expression, except for the dual RNA-seq study using the ex vivo whole blood model, for which it was possible to compare gene expression profiles between *C. auris* and other *Candida* species[19].

The absence of in vivo tissue infection profiling for *C. auris* may be due to the lack of adequate RNA recovery from current models. Most murine models, such as BALB/c[25,34–36] and C57BL/6 mice[24], have rarely demonstrated lethality through *C. auris* infection[23,37,38] without major immunosuppression with cyclophosphamide/cortisone[39–50], monoclonal antibody neutrophil depletion[26,51], or other immunodeficiency[27,52,53]. Additionally, mouse models are also expensive and associated with concerns over environmental pathogen shedding[54]. Alternative host models such as *Galleria mellonella*[34,44,55–61], *Drosophila melanogaster*[62] and *Caenorhabditis elegans*[60] have been employed, but these may lack key elements of mammalian-relevant immune systems, thus limiting their scope of application. In patients, immune control of *C. auris* is thought to depend on myeloid cell activity *via* C-type lectin receptors such as complement receptor 3 and mannose receptor MMR, followed by interleukin (IL-17) pathway activation[63,64]. The zebrafish (*Danio rerio*) model has allowed hindbrain[32,65] and swim bladder[66] microinjection modelling of *C. auris* infection, but rarely at temperatures above 30 °C, which zebrafish are broadly intolerant of. Fungal transcriptional programmes can vary greatly across temperatures and are critical for mammalian pathogenesis[67–69]. Despite these successes in profiling host responses, none of these models has enabled the profiling of *C. auris* RNA during infection.

A marine model could also shed light not only on human infection but also on a possible evolutionarily adaptive niche. The environmental origins of *C. auris* have not been conclusively demonstrated; however, a marine origin has been hypothesised and supported by the successful isolation of *C. auris* in coastal waters of both the Indian and Pacific oceans[70,71]. Notably, a close relative of *C. auris* in the *Metschnikowiaceae* clade, *C. haemulonii*, was first isolated in 1961 from the gut of blue-striped grunt (genera name "*Haemulon*" for their blood-red mouth interior)[72]. Fish larvae must survive hostile and varied ocean environments, demonstrating very early innate immune responses such as antimicrobial peptide production and the complement cascade[73,74]. Only in adulthood are teleost fish expected to possess functional IL-17 subtypes (A and F) that lead to T-helper-17 subset differentiation and adaptive immunity in response to fungal infection[75]. A thermotolerant teleost fish embryo model, such as the *Aphanius dispar*, also known as the Arabian killifish (AK), which can acclimatise to temperatures up to and including 40 °C[76–78], offers a balance between considerations such as suitability for experimental manipulation and applicability to human fungal infection at mammalian temperatures[79,80]. Additionally, fish embryo models align with the principles of replacement, refinement and reduction of animals in laboratory research and offer an opportunity to develop novel animal models with ethical benefits over models such as adult mice[81].

In this paper, we present the first study exploring the in-host virulence of *C. auris* clades I-V in the AK yolk-sac microinjection model at 37 °C. We supplemented our validation of the model with histology, fungal burden recovery, and dual host-pathogen RNA-seq. By comparing in vivo and in vitro expression profiles of *C. auris*, we detected a cross-clade expression signature that was significantly enriched for siderophore transporter *SIT1* paralogues, which mediate uptake of siderophores produced by other microbes (xenosiderophores). We designated this expanded gene family as

xenosiderophore transporter candidates (*XTC*), which included a haem transport-related (*HTR*) sub-clade. We also identified in vivo filamentation by clade V during infection, which was associated with orthologues of key hypha-associated genes from *C. albicans*. Together, these findings substantially enhance our knowledge of the processes employed by *C. auris* during infection.

## Results

### Modelling *C. auris* infection in *A. dispar*

AK yolk-sac microinjections with each of the five *C. auris* clades demonstrated lethality (defined by cessation of heartbeat) at 37 °C (92–100% vs 6% for sham injections; Fig. 1A, B). Earlier lethality was significantly higher for clade I compared to clades II ($p = 1.69^{-04}$), III ($p = 7.83^{04}$) or V ($p = 1.15^{03}$), and for clade IV compared to clades II ($p = 7.66^{03}$) or ($p =$ III $3.06^{-02}$, Fig. 1B, Data S1); median survival was lower for clade I and IV (70.4 and 92.9 h) *vs* clade II, III and V (106.7, 116.5, and 116.1 h). To investigate morphological outcomes of infection, we compared histological sections of infected yolk-sacs at 48 h post-infection (HPI). All clades exhibited budding yeast morphologies both in vitro and in vivo, except for the clade V strain, which demonstrated predominantly filamentous forms when grown in vivo (Fig. 1C). Growth kinetics showed only minor inter-clade variation. Clade II grew more slowly in YPD-broth at 37 °C (Fig. S1A), although this did not translate into lower CFU recovery from infected embryos (Fig. 1D). On average ($\bar{x}$), 673 CFUs were recovered per infected embryo post injection, which increased 166-fold at 24 HPI ($\bar{x} = 1.12 \times 10^5$) and by an additional ~60% at 48 HPI ($\bar{x} = 1.78 \times 10^5$). Filamentous clade V isolates formed rough (*vs* smooth) colonies on YPD-agar in approximately 1–2% of cases (Fig. S1B), composed of elongated yeast cells, filamentous and pseudohyphal structures measuring >10–50 μm (Fig. S1C). These results confirm that AK yolk-sac microinjection supports active *C. auris* infection, leading to embryo mortality by 7 days post-injection. This system therefore provides a robust and informative in vivo model for studying the pathogenicity of human fungal pathogens such as *C. auris*.

### Host responses to *C. auris* infection

To characterise host responses to *C. auris* during infection, we identified host differentially expressed genes (DEGs) in AK yolk-sacs between infection by each clade and sham injections (Fig. 2). Across all infections, 685 host DEGs were detected (representing 2.49% of potential genes), including 183 up-regulated and 508 down-regulated genes in response to at least one clade. Notably, there were no DEGs common to infection by all five clades. Only five DEGs (four up-regulated genes and one down-regulated gene) were shared across infections with four clades (Data S2). Up-regulated genes included Laminin subunit alpha-1 (*LAMA1*, not significant in response to clade III) and three genes that were not significant in response to clade II, Glucose-6-phosphatase (*G6PC*), DN117939 (hypothetical protein) and Tryptophan aminotransferase protein 1 (*TAR1*); immunoglobulin-like fibronectin type III domain-containing protein 1 (*IGFN1*) was down-regulated. These findings suggest host transcriptional responses to *C. auris* are modest and clade-specific at these early time points, possibly reflecting variation in virulence and stages of infection.

Although we did not observe a shared host transcriptional response across all *C. auris* clades, we identified distinct, clade-specific host responses to infection. At 24 HPI, clade II elicited the largest host response, with 41 up-regulated genes and 210 down-regulated genes (Fig. 3A). Minimal differences were detected between sham injection and uninjected controls, confirming the specificity of the infection-induced response (Fig. 3B, SI Appendix). By 48 HPI, clade IV induced the strongest transcriptional response, with 49 up-regulated and 104 down-regulated host genes. Notably, we detected differential expression of several key immune-related genes across clades, with the most pronounced changes in clades I and IV. These included heat shock proteins (*HSP30* and *HSP70*), complement cascade elements, pattern recognition receptors (PRRs), inflammatory transcription regulators, and genes involved in nutritional immunity (Fig. 3C). Among genes associated with pathogen recognition, *CARD14* was most

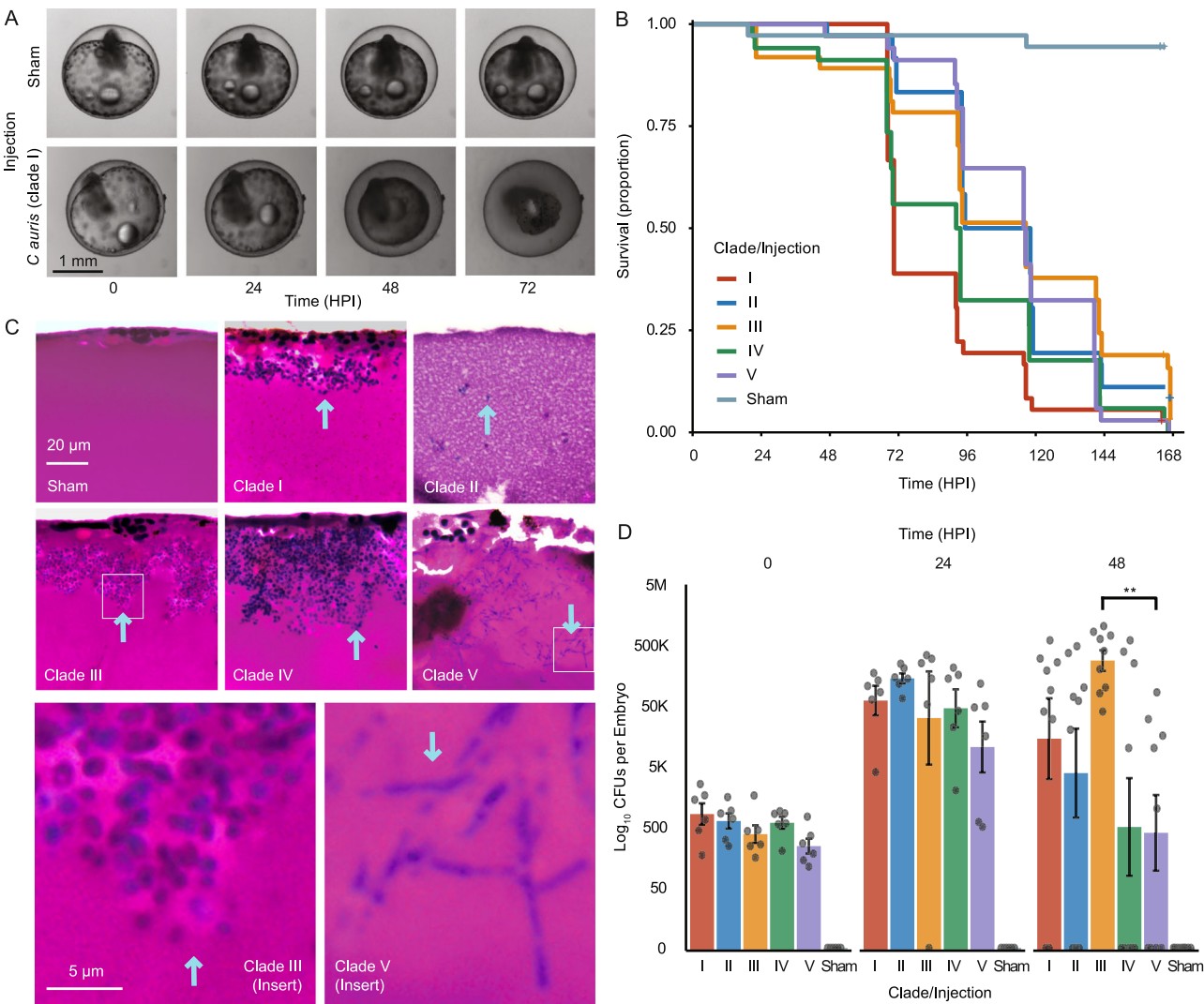

**Fig. 1 | Modelling *C. auris* infection in the Arabian killifish. A** Illustrative time-lapse microscopy of embryo death, including yolk-sac collapse during infection (clade I infection) or survival (sham injection) using Acquifer live imaging. **B** Survival curves for embryos injected with each *C. auris* clade or sham injection. Significance testing between groups is given in Data S1; hours post-injection (HPI). **C** H&E Staining of AK yolk-sac at 48 HPI for each clade or sham injection; morphology is indicated by blue arrows, with insets indicating typical yeast (clade III) and filamentous morphology (clade V). The yolk-sac edge is oriented at the top. **D** Colony forming units (CFUs) per clade from embryos homogenised after injection as an indicator of injection dose and in vivo growth.

prominently up-regulated during clade I and IV infection at 48 HPI (Fig. 3D, E), while *NLRP3* (a key inflammasome component) was expressed during clade III infection at 24 HPI.

Functional enrichment analysis of host DEGs revealed several significantly enriched Gene Ontology (GO) terms (and additional annotations), particularly in response to the more virulent clades I and IV at 48 HPI. Clade I infection was associated with 15 enriched GO terms, driven in part by up-regulation of *LAMA1* and *LAMA2* (orthologues of Laminin subunit-α, Fig. 3D). In clade IV-infected yolk-sacs, enriched terms were related to haem catabolism, haem binding, and haem oxygenase (*HMOX*) activity. Two *HMOX* orthologues were major contributors to this response, such as haem oxidation (log-fold change $\bar{x} = 3.96$, FDR < 0.0003, Fig. 3E, Fig. S3). Quantitative PCR confirmed increased expression of *HMOX* genes in the host; embryos infected by clade IV demonstrated late up-regulation at 48 HPI with a fold change of 3.26 and 9.21 for two *HMOX* genes, DN109585 and DN112160, respectively (compared to uninfected embryos; Fig. S2A). Another host up-regulated gene from infection by clade IV was ferroxidase hephaestin (*HPHL1*), which is also involved in iron nutritional immunity. Together, these findings suggest that heat shock proteins, nutritional immunity (particularly iron), and the complement effect system are important elements of the AK larval response to *C. auris* microinjection at 37 °C.

**Pathogen gene expression during *A. dispar* infection**

We compared the in-host transcriptome to expression induced by growth in a standard rich laboratory media (YPD). Principle components and sample correlation clustering from gene expression data indicated a predominant separate clustering of in vivo *vs* in vitro conditions, indicating that the gene expression profile of *C. auris* during infection is substantially different to that when it is grown in nutrient-rich laboratory media (Fig. S1G, S1H). We identified 960 DEGs between in vivo AK infection at 24 HPI or 48 HPI vs in vitro culture in YPD over 24 h (499 up-regulated, 523 down-regulated, 17.7% of potential genes, Fig. 4A). We identified few DEGs between the two time-points 24 HPI to 48 HPI, with zero DEGs for clade IV and V between those time points (Fig. 5A). The highest numbers of up-regulated DEGs were identified in the filamentous clade V at 24 HPI and in the more virulent clades I and IV at 48 HPI (Fig. 5B, C). Six functional annotation terms were enriched across all in-host infections at 48 HPI, related to major superfacilitator family

**Fig. 2 | Indicative pipeline diagram for methods used.** At 3 days post fertilisation (DPF), *A. dispar* were microinjected with *C. auris* from each of five clades I–V. Transcriptional comparisons were made between host and pathogen conditions using either a shared or clade-specific reference. Functional annotations were generated to ascertain enriched features. Illustrations © the authors.

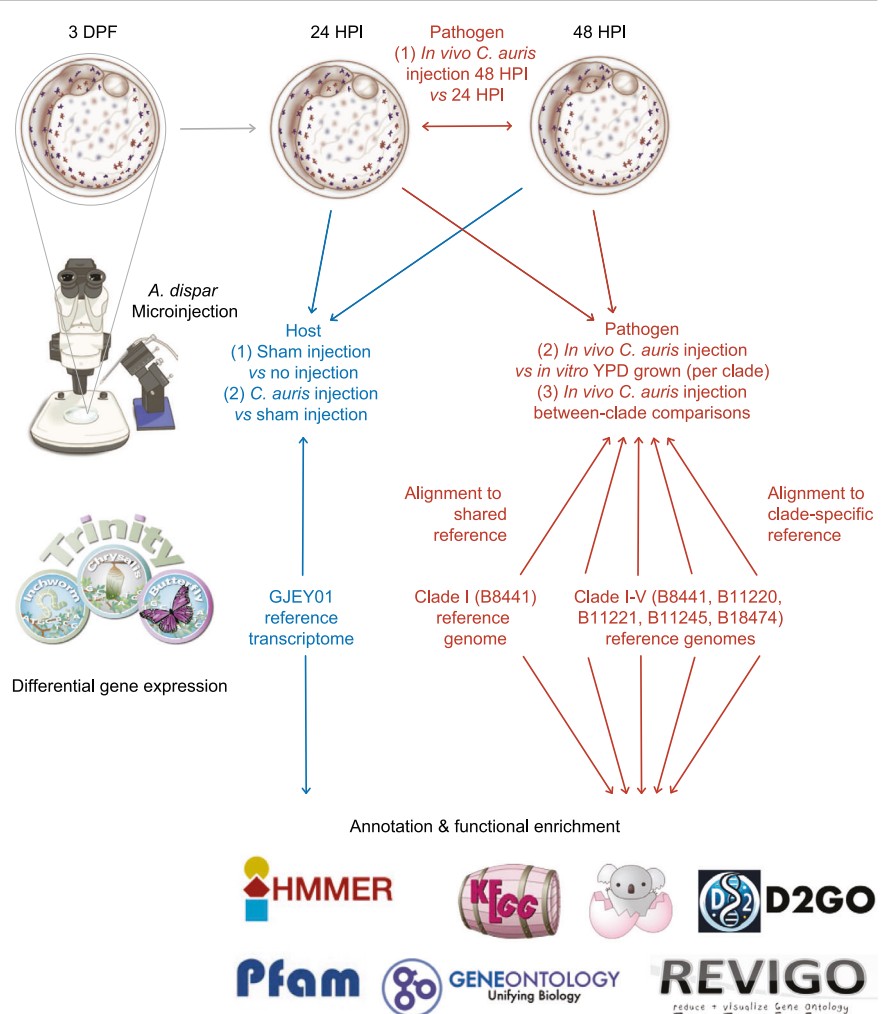

carbohydrate and siderophore-iron transmembrane transport (Fig. 6). Quantitative PCR confirmed increased expression of *SIT* genes that had been significantly up-regulated by *C. auris* at 48 HPI (Fig. S2B): during embryo infection, clade IV demonstrated up-regulation with a fold change of 2.55 and 8.16 at 24 and 48 HPI, respectively, for *XTC3* (B9J08_003921) compared to culture in YPD alone. Furthermore, *XTC7* (B9J08_001487) was up-regulated with a fold change of 39.5 and 130.8 at 24 and 48 HPI, respectively. Together, this analysis reveals that during infection, which is a nutrient-limited environment, *C. auris* is responding by significantly increasing expression of genes involved in sugar uptake, as well as members of the *C. auris* expanded gene family of siderophore transporters.

Thirteen DEGs common to all clades of in-host transcription were transmembrane transporters, including up-regulation of transporters for xenosiderophores (*SIT1* genes 3921 and 1487), sugar transporters (*HGT2*, *HGT12*), an ammonia transporter (*ATO2*), and a nicotinic acid transporter (*TNA1*). We also found down-regulation of several drug-resistance-related efflux pumps (*MDR2* and *DTR1*) and peptide transporters (*TPO3*, *PTR22*, Fig. 4B and Data S3). Surprisingly, several putative virulence factors such as lipase *LIP1* and cell wall-related *KRE6* and *SOD6* were also down-regulated across all five clades, as were *ALS4* and *MDR1* in at least one clade (Fig. 4C). Several members of the *SIT1* family were differentially expressed by <5 of the clades including 0002, 1457, 1458, 1542,2110, 2241, 2465, 3908, and 4097. Meanwhile, two *SIT1* genes were down-regulated across 3 of the clades (1519 and 1499). Several *SIT1* genes were upregulated in ≥1 clade and down-regulated in ≥1 clade (3908 and 2465). The strong signal of

differential expression in the *SIT1* expanded gene family highlights the potential importance during in-host survival.

**Transcriptional differences underlying virulence and filamentation**

The more virulent clades (I and IV) are **a**-type mating type locus (*MTL*), while the less virulent clades (II, III and V) contain the α-type *MTL*. Surprisingly, we found differential expression in several non-mating genes between more virulent and less virulent clades, including poly-A polymerase (*PAP1*), oxysterol binding protein (*OBPA*) and phosphatidylinositol 4-kinase (*PIKA*, Fig. 7B). Within the set of 10 DEGs shared across all comparisons, several genes outside the *MTL* were up-regulated, including vacuolar protein sorting gene, *VPS70*, and filamentous growth regulator, *FGR14*. Additionally, the drug efflux pump, *MDR1*, was down-regulated across all comparisons, supporting the idea of a fitness trade-off between resistance and virulence.

To explore drivers of filamentation by the clade V strain, we identified 1046 DEGs between clade V and each of the other clades (I-IV), comprising 601 up-regulated and 526 down-regulated genes across all comparisons. We identified a set of 57 genes shared across all comparisons for in vivo conditions only, since we did not observe filamentation in nutrient-rich YPD. Twenty-two genes were up-regulated at 24 HPI, seven at 48 HPI, and 18 at both time-points (Fig. 7C). Most of the up-regulated genes were metabolic (*n* = 13), including two cytochrome P450 *ERG5* orthologues, though *ERG11* was down-regulated. Transmembrane transporters for sugar (including two *HGT13* orthologues), amino acids and siderophores (*SIT1* 1487) were also up-regulated (*n* = 10). Cell wall genes up-regulated in filamentous clade V

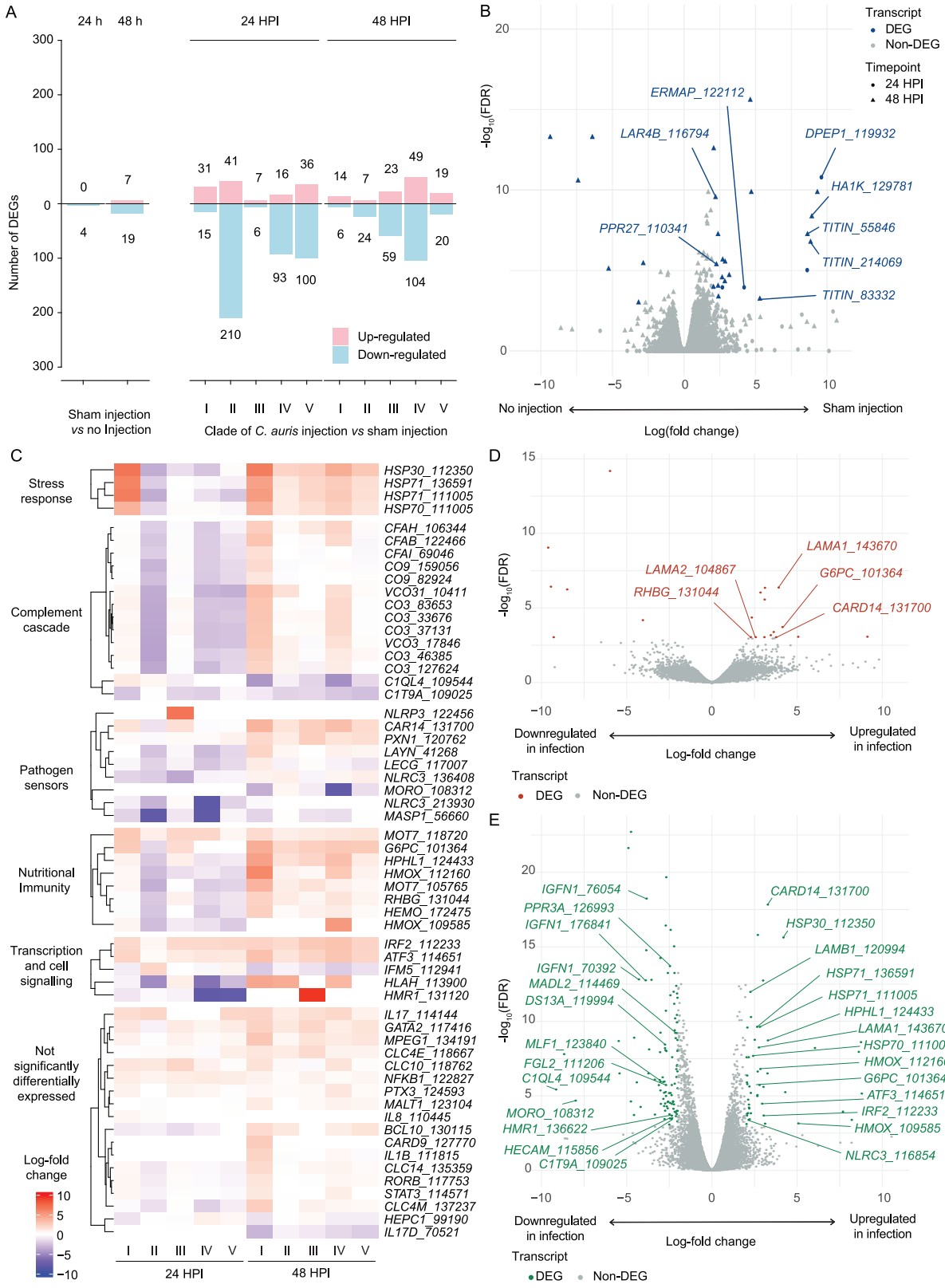

**Fig. 3 | AK host responses to *C. auris* infection. A** Number of differentially expressed genes (DEGs) for each of two comparisons, sham injection *vs* no injection (left panel), and *C. auris* injection *vs* sham injection (right panel), at either 24 or 48 h post-injection (HPI). **B** Volcano plot showing log-fold change and significance according to -log[10] false discovery rate (FDR) for sham injection *vs* no injection.

**C** Heatmap demonstrating log-fold change for differential expression between *C. auris* injection and sham injection for groups of genes. **D** Volcano plot showing log-fold change and significance for DEGs expressed during infection with clade I *C. auris* at 48 HPI. **E** Volcano plot showing log-fold change and significance for DEGs expressed during infection with clade IV *C. auris* at 48 HPI.

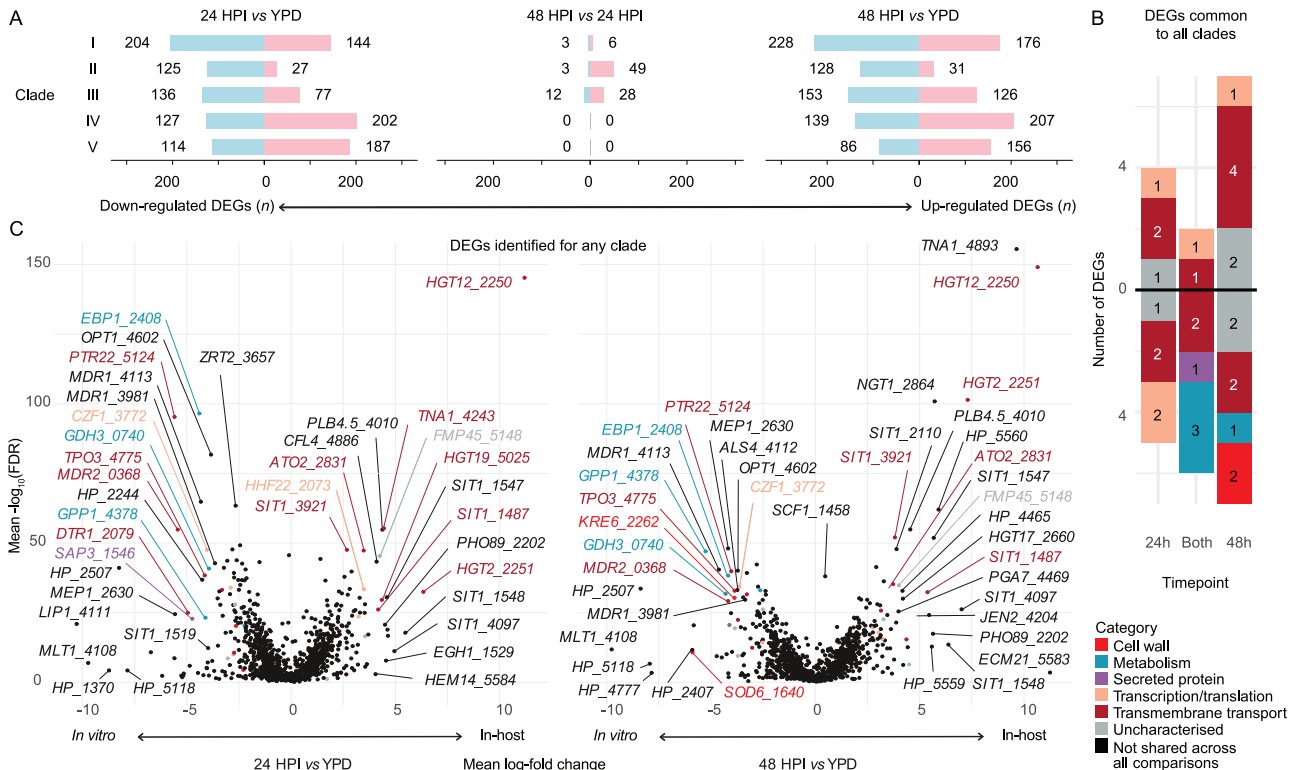

**Fig. 4 | *C. auris* gene expression *in vivo vs in vitro*. A** Total DEGs per clade for each comparison between 24 h post injection (HPI), 48 HPI, and growth in YPD alone. **B** Combined set of significant DEGs common to all five clades between in vivo infection and in vivo growth in YPD. Genes are coloured by manually annotated category (Data S3). **C** Volcano plots demonstrate mean log-fold change for DEGs identified for any clade. Genes with infinite false discovery rate (FDR) are indicated with an arrow. Genes included in the combined set are coloured by category as per (**B**).

($n$ = 8) included the novel adhesin *SCF1*, GPI-anchored genes *PGA31* and *IFF4*, and orthologues of classical *C. albicans* hyphal adhesins *ALS3* and *RBR3*. We identified differentially expressed transcription factors and related regulators, such as *HAP41* and the pheromone receptor *STE2*, which could play a governing role on these responses.

We also examined significant DEGs between clade V and at least one other clade using mean-log fold change (Fig. 7D). Several genes that play critical roles in *C. albicans* hypha formation were highly up-regulated, such as transcription factor *UME6* and hypha-specific G1 cyclin-dependent protein serine/threonine kinase *HGC1*. Enrichment analysis indicated up-regulated gene sets involving CFEM domain, cytochrome P450/oxidoreductase activity, hypha-regulated cell wall GPI-anchored proteins and siderophore transport, particularly at 24 HPI and comparing clade V to clade II (Fig. 7D). Significantly up-regulated iron-related transport genes in clade V included *FRE3*, *CFL4*, *FRP1*, and *SIT1* genes 1487, 1499, 1519, 1542, and 4097. This is consistent with the preservation of a major filamentation programme in *C. auris*, demonstrated in the basal clade V.

**Xenosiderophore transporter candidate (*XTC*) and haem-transport related (*HTR*) expanded gene family**
Given the prominence of *SIT1* orthologues across the transcriptomic analyses of *C. auris* during in vivo gene expression, we set out to characterise the structural and evolutionary conservation of the expanded gene family across orthogroups, including outgroup species *C. haemulonii* and *C. albicans* (Note S1). We performed phylogenetic analysis on proteins with sequence similarity across seven orthologue clusters, of which two main groupings of transporters with 14 transmembrane domains (TMRs) emerged (Fig. S5). The first group included genes with functional annotation relating to siderophore import (including 12 genes in clade I). The second group included haem transmembrane import genes (including 5 genes in clade I). Most of the orthologues retained 14 TMRs, and a few had evidence of additional annotations except for the trichothecene mycotoxin efflux pump

PFAM domain across multiple groupings. The orthologue *SIT1* 1547 contained 12 TMRs and was up-regulated with *SIT1* 1548 containing 2 TMRs, which was not identified as a *SIT1* orthologue in our pipeline but has previously been described as such[19,82].

To explore the evolutionary history of *SIT1*-related genes in fungi, we performed Blastp searches for orthologues of *C. albicans SIT1* using an e-value cut-off of $1e^{-20}$ across 226 fungal species' reference genomes ($n$ = 304). We identified 1689 hits, which clustered into two branches: a smaller branch ($n$ = 397) annotated for siderophore transport, including the 12 *C. auris* groupings which we have designated "xenosiderophore transporter candidates" (*XTC*), and a larger branch ($n$ = 1292) featuring genes predicted (by GO term) to encode haem transporters, including the 5 *C. auris* groupings, which we have additionally designated "haem transport-related" (*HTR*, Fig. 8A). There were no putative haem transporters or members of the larger branch for *C. albicans*, which only contained the single *SIT1* gene. We calculated the mean number of orthogroups per genome per species (Fig. 8B). This showed the most prominent expansion of the *XTC* branch in the *Metschnikowiaceae* clade, including *C. auris* (12 orthologues) and even more so in *C. haemulonii*, with 15 orthologues. No other fungal species or group of species contained so many. *XTC* orthologues were not present in any members of the basal *Blasocladiomycota*, *Chytridiomycota* and *Mucoromycota*, or in several *Ascomycota* (e.g. *Pichia kudriavzevii*, *Schizosaccharomyces pombe* and *Pyricularia oryzae*) and Basidiomycota (e.g. *Ustilago maydis*). Common human fungal pathogens such as *Aspergillus fumigatus*, *C. albicans* and *Nakaseomyces glabratus* contained only one *XTC* orthologue (Fig. 8B). The *HTR* clade, by contrast, was more prominent across fungal phyla, with as many as nine in *Mucor* species, 16 in *A. flavus* and 22 in *F. solani*. We therefore propose a naming of these two expanded gene families as *XTC1-12* and *HTR1-5* in *C. auris* (Table 1).

We note the absence of *XTC5* in clade V, *XTC10* in clade IV, and in addition to these, *XTC4*, *XTC9* and *XTC11* in clade II (Fig. 8C). Clade II,

**Fig. 5 | *C. auris* gene expression in vivo vs in vitro.**
**A–C** Upset plots for each in vivo vs in vitro comparison between 24 h post-injection (HPI), 48 HPI, and growth in YPD alone, indicating DEGs that were significant across all five clades (far left) and for each individual clade (far right) and each set in between.

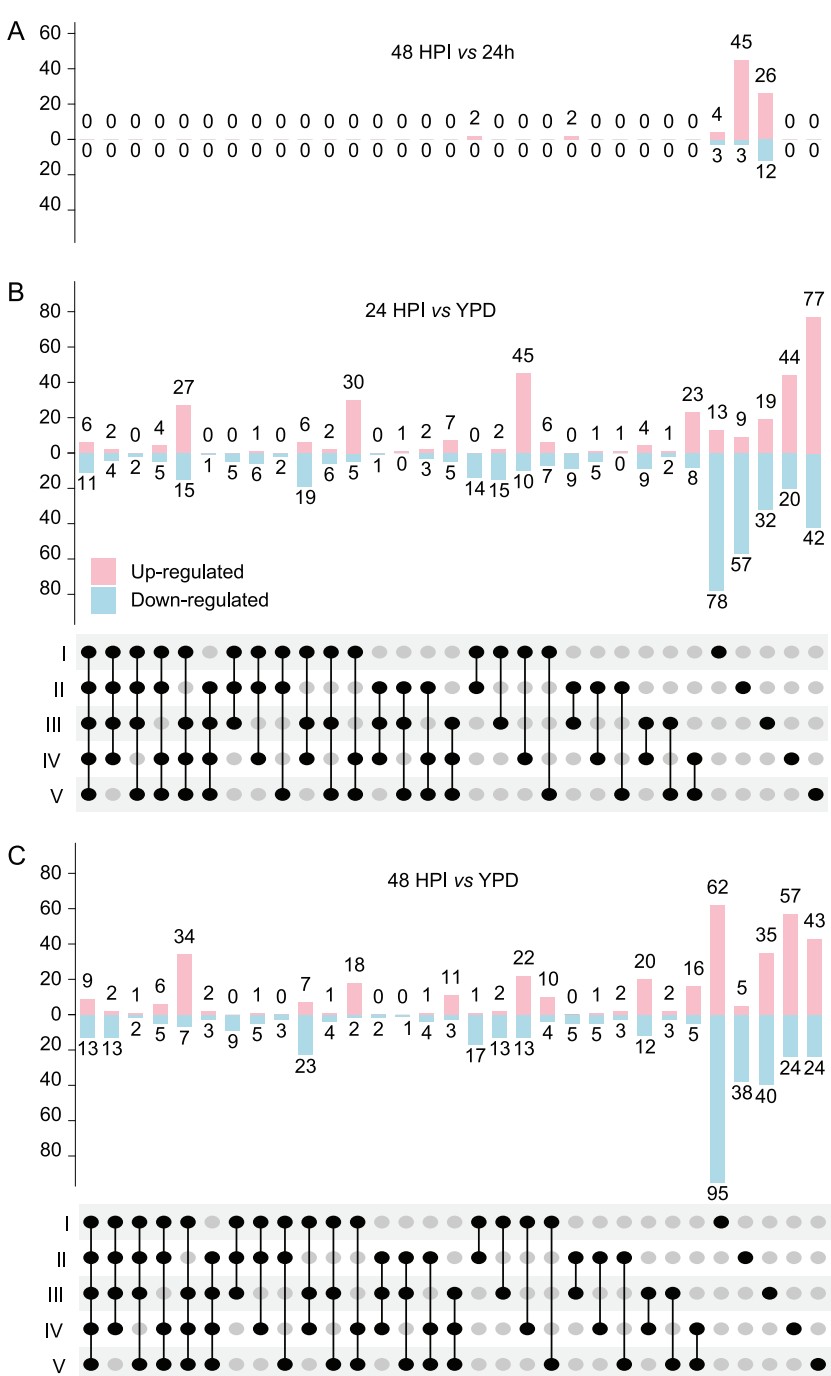

which exhibits the greatest degree of *XTC* gene loss, is also considered by some to be the least virulent strain. It is also noteworthy that the AK embryo expressed *HMOX* genes during clade IV infection, while clade IV *C. auris* up-regulated *HTR1*, *HTR2* and *HTR5*. This suggests a specific host-pathogen interaction axis in nutritional immunity related to haem scavenging. No other clades of infection were associated with such iron-related functional enrichment in the host, and no other clades expressed more than one *HTR* gene. In summary, *XTC1*, *XTC2*, *XTC3*, *XTC4*, *XTC6*, *XTC7*, *XTC11* and *XTC12* were up-regulated in-host, while *XTC8*, *XTC9* and *XTC12* were down-regulated in at least one clade, while up-regulation of *XTC3* and *XTC7* was significant across all clades. Additionally, the more virulent clades were associated with up-regulation of *XTC4*, *XTC5* and *XTC8*. The filamentous clade V up-regulated *XTC7* across comparisons with all other clades, and up-regulated *XTC1*, *XTC4*, *XTC8* and *XTC9* compared to at least one other clade.

## Discussion

We describe the use of an AK yolk-sac microinjection model for comparative functional genomics of *C. auris* during in vivo in-host infection. All five clades demonstrated lethality, with an increase in virulence during clades I and IV infections, consistent with clinical and laboratory findings[17]. In-host growth was similar across all clades and high enough to indicate a large enough in vivo fungal burden for robust bulk transcriptome interrogation across the species. The contained nature of the yolk-sac provides a host tissue growth medium that enables the assessment of pathogen survival and virulence strategies in vivo and a straightforward experimental unit of

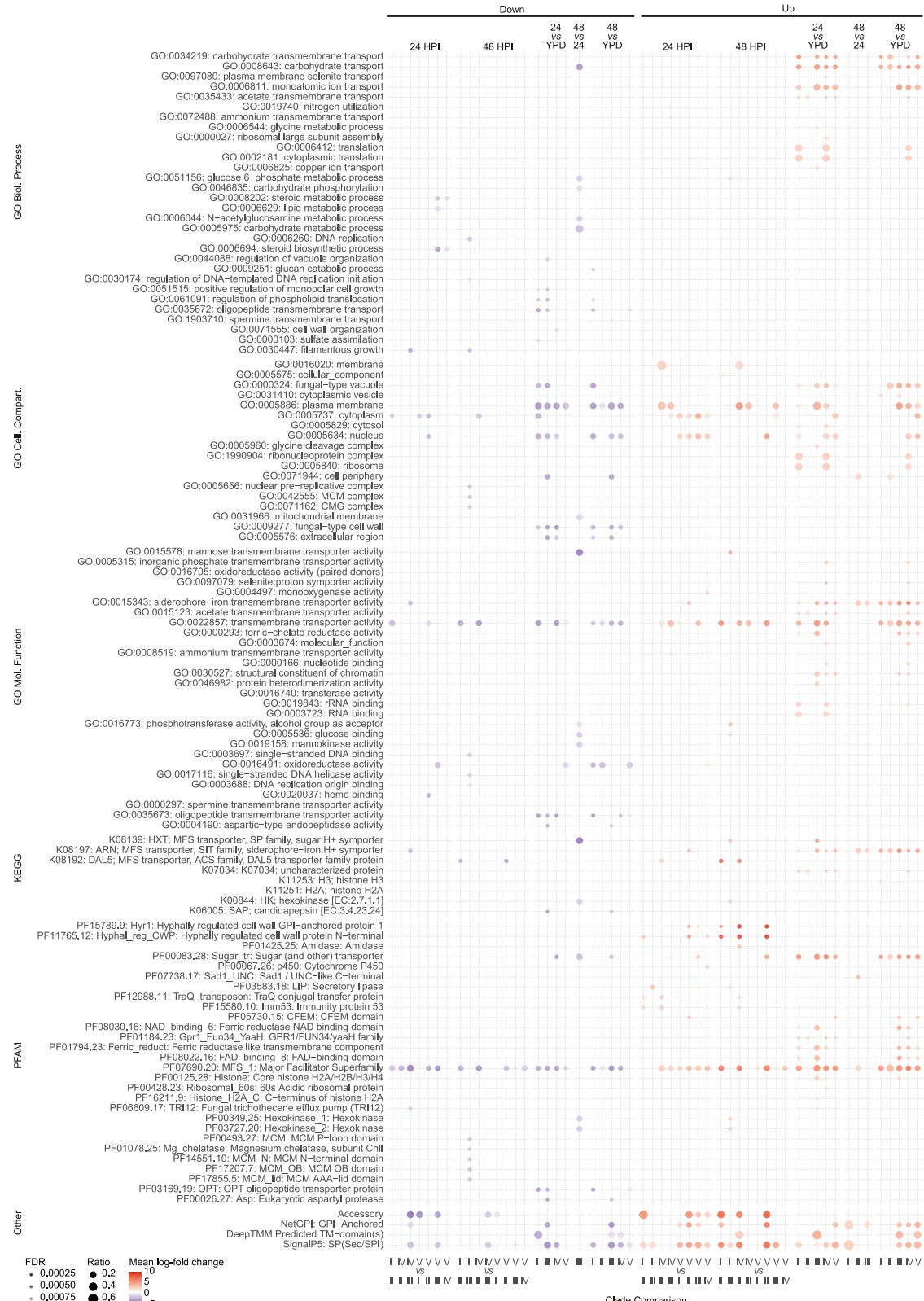

**Fig. 6 | *C. auris* gene set, pathway and domain enrichment:** Transcripts from *C. auris* reference genomes for each clade were annotated for Gene Ontology (GO) terms, Kyoto Encyopaedia of Genes and Genome pathways (KEGG), PFAM domains, transmembrane domains, GPI-anchored domains, SignalP domains, and membership within the accessory genome. For each set of up-regulated or down-regulated genes in each comparison, Fisher's exact test was used with Benjamini–Hochberg (BH) correction for multiple testing for each set, with a false discovery rate (FDR) cut-off of 0.001. Mean log-fold change and ratio of genes in that set were calculated for genes in each set possessing each feature.

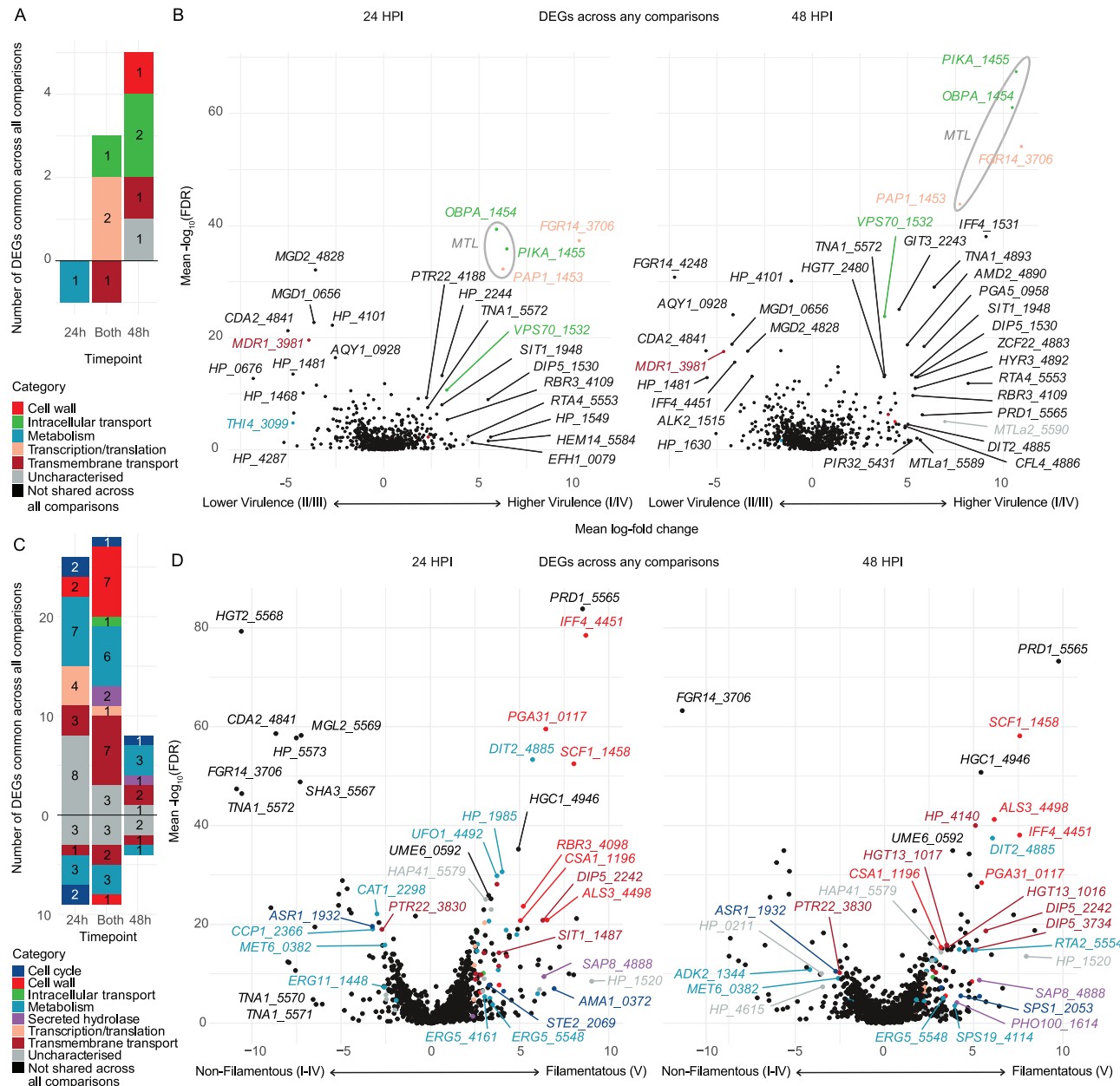

**Fig. 7 | Interclade comparisons for virulence and filamentation. A** The set of differentially expressed genes (DEGs) shared across all comparisons between clade I and IV (more virulent) and clade II and III (less virulent), coloured by manually annotated category, at either 24 or 48 h post infection (HPI). **B** Volcano plots demonstrate mean log-fold change for each DEG found in any comparison; genes included in the combined set are coloured as per (**A**). **C** The set of DEGs shared across all comparisons between filamenting clade V and all other clades (I–IV) in vivo, coloured by manually annotated category. **D** Volcano plots demonstrating mean log-fold change for all comparisons between clade V and clades I–IV at 24 HPI and 48 HPI for each DEG found in any comparison; genes included in the combined set are coloured as per (**C**).

host tissue without the challenges of pathogen RNA isolation, such as locating foci of infection in larger animals. Thus, an AK yolk-sac micro-injection model has the potential to fill a substantial need for in vivo gene expression studies in general.

We investigated the host transcriptional response to *C. auris* infection, which featured an innate immune response, as expected. The most highly up-regulated genes included two predicted haem oxygenase (*HMOX*) genes. *HMOX* genes have been shown to mask iron from invading pathogens as a feature of anti-*Candida* nutritional immunity[83], so presumably have a similar function against *C. auris*. This was particularly notable given *HMOX* genes were up-regulated in response to clade IV infection, and the clade IV isolate had a

corresponding significant up-regulation of three haem transport-related *HTR* genes. In addition to nutritional immunity genes, we found up-regulated heat shock and complement cascade genes during infection, confirming the expected immune response to fungal infection[63,64]. Genes with complex roles in immunity were up-regulated, such as *NLRP3*, for which *C. auris* may be less activating than e.g. *C. albicans*[33,84]. It is notable that the yolk-sac model was not associated with histological or transcriptional evidence of host neutrophil recruitment and activation, which may relate to the hypothesis that *C. auris* is immunoevasive[22]. Reference genome bias resulting from the available adult gill transcriptome assembly[85] may limit detection of host gene expression features, so future long and

**Fig. 8 | Xenosiderophore transporter gene evolution across the fungal kingdom and differential expression across clades in the Arabian killifish. A** Panfungal xenosiderophore phylogeny based on blastp hits for *C. albicans SIT1* across fungal genomes available in FungiDB. Sequences were aligned with Clustal Omega and FastTreee, scale bar: substitutions per site, *XTC*: xenosiderophore transport candidate, *HTR*: haem transport-related. **B** Per-species mean Blastp hits for siderophore transporters. **C** Differential expression of *XTC/HTR* genes assigned in order of B8441 v3 contig across experiments in this paper.

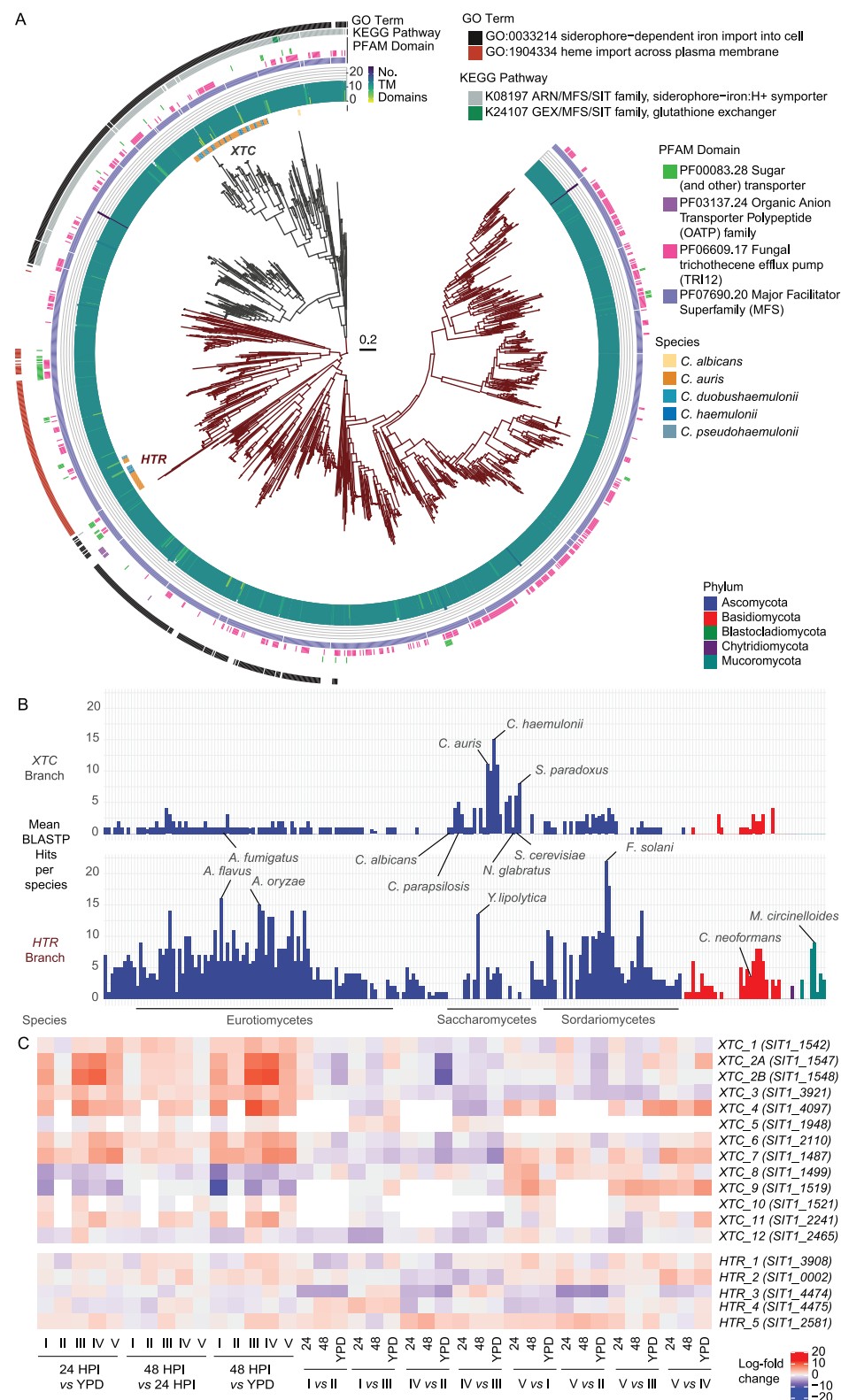

short read assembly from gene expression data across a range of larval stages could further understanding of AK immunity. Additional time-points could be an informative future area of study to develop fine-resolution strain-specific transcription networks. Furthermore, the potential of a translucent and genetically tractable fish embryo model could enable future visualisation of cellular dynamics and genetic pathway manipulation[77].

We provided a thorough investigation of differential expression by *C. auris* during infection of host tissue compared with in vitro conditions. The lack of a major distinction across time-points (24 *vs* 48 HPI) indicates that

**Table 1 | Summary of *XTC* and *HTR* genes and their chromosomal locations**

| Proposed name | Shortened name | B9441 v2 code | B8441 v3 code | B8441 v3 contig | Chromosome |
|---|---|---|---|---|---|
| XTC_1 | SIT1_1542 | B9J08_001542 | B9J08_00013 | CM076438.1 | 1 |
| XTC_2A | SIT1_1547 | B9J08_001547 | B9J08_00018 | CM076438.1 | 1 |
| XTC_2B | SIT1_1548 | B9J08_001548 | B9J08_00019 | CM076438.1 | 1 |
| XTC_3 | SIT1_3921 | B9J08_003921 | B9J08_01779 | CM076438.1 | 1 |
| XTC_4 | SIT1_4097 | B9J08_004097 | B9J08_01954 | CM076438.1 | 1 |
| XTC_5 | SIT1_1948 | B9J08_001948 | B9J08_03021 | CM076440.1 | 3 |
| XTC_6 | SIT1_2110 | B9J08_002110 | B9J08_03183 | CM076440.1 | 3 |
| XTC_7 | SIT1_1487 | B9J08_001487 | B9J08_03737 | CM076440.1 | 3 |
| XTC_8 | SIT1_1499 | B9J08_001499 | B9J08_03749 | CM076440.1 | 3 |
| XTC_9 | SIT1_1519 | B9J08_001519 | B9J08_03769 | CM076440.1 | 3 |
| XTC_10 | SIT1_1521 | B9J08_001521 | B9J08_03771 | CM076440.1 | 3 |
| XTC_11 | SIT1_2241 | B9J08_002241 | B9J08_05251 | CM076444.1 | 7 |
| XTC_12 | SIT1_2465 | B9J08_002465 | B9J08_05475 | CM076444.1 | 7 |
| HTR_1 | SIT1_3908 | B9J08_003908 | B9J08_01766 | CM076438.1 | 1 |
| HTR_2 | SIT1_0002 | B9J08_000002 | B9J08_01961 | CM076439.1 | 2 |
| HTR_3 | SIT1_4474 | B9J08_004474 | B9J08_04433 | CM076442.1 | 5 |
| HTR_4 | SIT1_4475 | B9J08_004475 | B9J08_04434 | CM076442.1 | 5 |
| HTR_5 | SIT1_2581 | B9J08_002581 | B9J08_05591 | CM076444.1 | 7 |

the transcriptional activity of *C. auris* in-host was highly similar at these stages. Six functional annotation terms (GO, KEGG pathway and PFAM domain) were significantly enriched among genes that were up-regulated across all five clades, including sugar and siderophore transporters. The enrichment of sugar transporters outside of nutrient-rich YPD-broth is unsurprising, such as high-affinity glucose transporter *HGT12*, with evidence of redundancy in *C. albicans*[86–89], and *HGT2*, part of an eight-gene 'core filamentation response network' in *C. albicans*[90]. More surprising was the down-regulation of putative virulence factors such as Secreted Aspartyl Protease 3 (*SAP3*), thought to be the most important SAP in *C. auris*, and required for virulence in a mouse model[31]. Copper Superoxide Dismutase *SOD6* was also down-regulated, previously shown to be down-regulated in iron starvation[91]. Multi-drug efflux pumps such as *MDR2* (down-regulated in vivo) and *MDR1* (down-regulated in vivo and in clades I and IV *vs* clades II and III) are understood to drive drug resistance through multiple but unclear mechanisms in *C. auris*[92,93]; in-host down-regulation may indicate a fitness trade-off for virulence *vs* drug resistance. A total of 12 siderophore transporters were up-regulated during infection, and four were down-regulated, with three family members up-regulated in more virulent clades and five in the filamentous clade V. The prominent finding of these xeno-siderophore transport candidate (*XTC*) genes necessitates further experimental work to address their potential for anti-virulence therapeutic targeting.

We also identified up-regulation of several genes at the mating type locus (*MTL*) in more virulent strains; *MTL* genes may contribute to virulence in fungal species, including *Cryptococcus neoformans*[94], through uncertain mechanisms, such as *via* the expression of non-mating genes contained within the locus. *C. auris* does not clearly undergo meiotic recombination[95] and it is not clear what role *MTL*-associated genes play in pathogenesis and morphology. Two of the three non-mating genes (*PAP1* and *PIKA*) in the *MTL* were up-regulated by more virulent strains (I and IV) compared to the less virulent strains (II and III). Importantly, the two more virulent clades (I and IV) contain *MTL**a*** rather than *MTLα* (clades II & III)[29], and clades V and VI contain *MTL**a*** and *MTLα*, respectively[13]. Thus, it is unclear if the up-regulation of non-mating genes is a *MTL* specific transcriptional nuance or a feature of more highly virulent *C. auris* clades. The *MTL* also did not feature prominently in the filamentation response

observed in clade V, providing no evidence for an obvious role for the *MTL* in such morphological changes.

We also observed in vivo filamentation by a representative strain of *C. auris* clade V, the basal/ancestral clade[96], accompanied by a genera-typical filamentous gene expression signature. Specifically, we measured the up-regulation of *HGC1* and *UME6*, which are two of the most important regulators of filamentation in *C. albicans*, and have recently been described as drivers of *C. auris* filamentation and biofilm formation[97]. The expression of adhesin *SCF1* in filamenting clade V is noteworthy because its cationic adhesion mechanism has been postulated to be consistent with adhesion mechanisms common to marine organisms, including bivalves[27]. The generalist marine-isolated pathogen *C. haemulonii* has also displayed filamentation competence at lower temperatures[98], consistent with an ancestral aquatic state. Filamentation is an important pathogenicity trait in fungi, especially *Candida* species, although its roles have been more poorly described in aquatic infections[99].

Human histopathology reports of *C. auris* infection are rarely published and have not demonstrated classical in-host filamentation, to our knowledge. An ex vivo human epithelial model using a clade I isolate did demonstrate filamentous *C. auris* at 37 °C after injection into deeper dermis[100]. Clade III *C. auris* has also been observed to form filamentous/hyphal forms in *G. mellonella* injection[58]. Fan and colleagues have observed filamentous forms of *C. auris* across all four clades, even demonstrating a higher virulence in the *G. mellonella* model[101], though the phenotype required passage through a murine host with additional prolonged growth at 25 °C for at least 5 d before and after injection[25]. Other stimuli have induced filamentation in clade I isolates, such as hydroxyurea genotoxic stress[102] and Hsp90 inhibition[103], where several up-regulated genes in *C. auris* filamentation (*SCF1*, *IFF4* and *PGA31*) matched those in our study. Constitutive filamentous growth has also been demonstrated with a transposon-mediated disruption of a long non-coding RNA[104], which sit adjacent to chitin deacetylase *CDA2*, which was down-regulated at 24 HPI in our experiment during filamentation. These findings indicate that hyphal formation is possible across *C. auris* clades.

We underscored how the family of predicted xenosiderophore transporter genes is significantly expanded in several members of the *Metschnikowiaceae* clade compared to other *Saccharomycetae*[29], raising questions

as to the roles of siderophore transporters in *C. auris* biology and evolution[105]. Siderophore synthesis may be unlikely in *C. auris*, given the lack of evidence in other *Candida* species[106]. As a result, *C. albicans* Sit1, which is required for virulence[107], is thought to be a xenosiderophore transporter for iron-chelators produced by other microbes and specific to the ferrioxamine type common in iron-poor ocean environments[108,109]. Siderophore transporters can also act intracellularly for compartments such as the mitochondrion[110,111], or act as efflux pumps for various molecules[112], including trichothecene mycotoxins as identified by our functional annotations. Given that *XTC* gene up-regulation by *C. auris* has also been detected during ex vivo whole blood infection (*XTC2A-B*, *B9J08_1547-8*), this suggests that our findings may be clinically relevant[19].

*C. auris* encodes a large expanded gene family of 17 siderophore transporters that we named xenosiderophore transporter candidates (*XTC*) 1-12 and haem transport-related (*HTR*) genes 1-5, the latter sharing sequence similarity with the genomic sequence for *S. pombe* low-affinity haem transporter Str3[113]. In other *Candida* species, such as *C. albicans*, iron assimilation is understood to be central for in-host survival[114,115], where deprivation triggers various pathogenic programmes such as alterations in cell surface beta-glucan[116]. Haem is acquired by *C. albicans* through vesicular endocytosis and CFEM-domain containing proteins[117,118] and does not contain *HTR* genes, suggesting a possible loss of transmembrane haem transporters. The ubiquity of *HTR* across the fungal kingdom and lack of *XTC* genes in the three basal fungal phyla suggest that *SIT1/HTC* could have arisen from a duplication of an *HTR*-related gene in the *Dikarya* (the *Ascomycota* and *Basidiomycota*) with subsequent independent loss of both *XTC* and *HTC* genes in multiple fungal lineages. These findings together underline the importance of iron metabolism as a clinical target with existing therapies, such as iron chelators deferiprone/deferoxamine with evidence of echinocandin antifungal synergy[119] or low toxicity antihelminthic pyrvinium pamoate as a disruptor of iron homoeostasis[120,121].

Limitations to this study include the lack of gene knockout studies related to the expanded gene family of *XTC/HTR* transporters. Depicting the potentially overlapping roles in expanded gene families is a challenge requiring extensive bodies of research[122]. Another expanded gene family in *C. auris* is the Hil (*Hyr/Iff*-like) family, including eight members with similar motifs (effector domain, signal peptide and GPI-anchor) but differing tandem repeat section lengths, which may result in functional variation[123]. Given the unique expansion of *XTC* genes in the *Metschnikowiaceae* clade, assessment of the role in related pathogenic *Candida,* such as *C. haemulonii*, may shed light on species-specific adaptation *via* gene family expansion. Further gene knockouts could confirm roles of DEGs with potential roles in filamentation and virulence, supplemented by additional testing in potentially more clinically relevant models, such as ex vivo whole blood or in vivo murine virulence studies. Finally, investigating a potential antifungal role for manipulation of iron metabolism will require fundamental ascertainment of pharmacokinetics and pharmacodynamics of iron chelators in the yolk sac and their effects on fungal growth and gene expression. Development of the AK model is also required to image host immune cells with fluorescent fungal strains, improve reference genome databases, and determine the effects of traditional antifungal drugs on *C. auris* gene expression during infection.

We present an in vivo host tissue infection transcriptional profile for *C. auris*, a World Health Organization critical priority fungal pathogen. By examining representative strains from all five major clades implicated in its recent, near-simultaneous global emergence, we uncover clade-specific differences in morphology, virulence and gene expression. Our findings demonstrate the utility of a thermotolerant yolk-sac injection model and support the hypothesis of a marine origin for *C. auris*. Notably, we observed natural filamentation in clade V and elevated virulence in clades I and IV, each linked to distinct transcriptional programmes. We identify the expanded *XTC* and *HTR* families as species-wide, infection-induced features, underscoring their potential as targets for studying fungal biology, pathogenesis, and antifungal development. Together, our results shed light on the mechanisms by which *C. auris* adapts to and persists in host tissue, offering insights into its evolution from a likely marine yeast ancestor and

advancing our understanding of this emerging fungal pathogen and global public health threat.

## Methods

### *Aphanius dispar* care, monitoring, husbandry and procedures
Wild-type adult AK aged 18–36 months were kept at the Aquatic Resources Facility, Exeter, in two 28 °C incubation tanks (*n* = 18, 20), with male and female fish in a 1:2 ratio, 12-h day and night light cycles, and artificial enrichment foliage. Embryos were collected using three plastic netted collection chambers per tank with transfer of all enrichment into tanks, and placed overnight from 16:00 to 09:15 in dark conditions. Chambers were then drained, cleaned and strained with 35 parts per thousand artificial seawater (ASW) to isolate all embryos. Subsequent ASW was sterile suction filtered *via* 0.2 μm pore Nalgene Rapid-Flow (Thermo Scientific, UK). Embryos were inspected for viability in terms of an intact blastocyst without discolouration by stereo dissection microscope (Optech, UK) and incubated at 30 °C (Memmert, Germany) for 72 h with day and night cycles with daily ASW changes and removal of dead, unhealthy or underdeveloped embryos, with a maximum of 30 embryos per standard size 100 × 15 mm petri dish (Greiner Bio-One, Austria). Excluded embryos were killed or disposed of either by submission in Virkon solution (Sigma-Aldrich, UK) or freezing at −80 °C.

### Experimental procedures
Injection needles were prepared using borosilicate glass capillaries (Harvard apparatus, USA) and a heated pulling system (heater level >62, PC-10, Narishige, UK). We used tweezers to snap the tip to an estimated 1 mm graticule-measured aperture of 10–20 μm, and sterilised needles before use with 253.7 nm UV light over 15 min (Environmental Validation Solutions, UK). Injection bays were made with microwaved 1% agarose in 1 g (Sigma, USA) per 100 mL ASW for up to 24 embryos in 100 mm petri dishes using custom-made moulds in an inverted 60 mm petri dish. Embryos were randomly chosen from each dish, and the sequence of injection for each biological repeat was randomly allocated by computer. For microinjection, *C. auris* yeasts were counted by haemocytometer and added to 25 μL sterile filtered phenol red solution 0.5% (Sigma, UK) to a concentration of $0.5 \times 10^8$ per mL in 100 μL. Injection needles were loaded with 5 μL injection solution using 20 μL Microloader EpT.I.P.S. (Eppendorf, UK) immediately before injection of up to 30 embryos, before reloading. Injections were performed with 400–500 hPa injection pressure and 7–40 hPa compensation pressure over 100–200 ms with the FemtoJet 4i (Eppendorf UK) using a micromanipulator (World Precision Instruments, UK) and a Microtec HM-3 stereo dissection microscope (Optech, UK). In order to deliver a dose of approximately 500 yeasts in 10 nL solution, an injection bolus was administered to an estimated diameter of 0.2–0.3 mm (10–15% of total embryo diameter) *via* a single chorion puncture. Any embryos that failed to meet these criteria for injection were excluded. Control in vitro inoculations were microinjected as above into 1 mL YPD in 1.5 mL Eppendorf tubes before incubation without shaking at 37 °C for 24 h, contemporaneously with embryos as below.

### Statistics and reproducibility
For the calculation of sample size per group, we determined that ≥23 embryos per condition were required, based on a power calculation of $\sigma = 0.25$, significance = 0.05, power = 0.9, and $\delta$ effect size of 25% mortality (two-sided *t*-test). To demonstrate differences in virulence over biological triplicate experiments, Kaplan–Meier survival curves were plotted with Log-Rank testing and Benjamini–Hochberg (BH) correction[125] using survival v.3.5.8 and survminer v.0.4.9 packages in R v.4.4.0. RNA-seq experiments were based on DNA extracted from three separate embryos per condition as technical replicates. RNA was extracted separately for each embryo, except for two embryos (negative control embryos without injection at 24 HPI), which were pooled together and analysed as a single datapoint alongside a third embryo analysed as a duplicate. qPCR readings were from triplicate replicates per condition. Growth curves were taken from four colonies per strain in technical triplicate resulting in twelve data points per colony.

## Study design and sample sizes

We used an unblinded prospective virulence study comparing six groups of AK embryo yolk-sac microinjection, including clades I-V of *C. auris*, and sham injections as the negative control. After excluding embryos that were unhealthy or did not complete successful microinjection, we included $n = 10$–13 embryos per group per experiment (total $n = 34$–37 embryos per group). Each embryo was considered to be an experimental unit ($n = 70$–72 embryos per experiment, total $n = 213$ embryos). For estimation of *C. auris* in-host growth by homogenisation and plating of colony-forming units, three embryos per condition per time-point across two (0 and 24 HPI) or three (48 HPI) experiments were used (total $n = 126$). For time lapse microscopy demonstration of *A. dispar* yolk-sac collapse and death, representative image sets were chosen from a pilot experiment using a reduced dosage (1/10) of *C. auris* ($n = 4$). For histological demonstration of *C. auris* morphology, one embryo per condition at 24 HPI and 48 HPI each were used in one experiment, and seven embryos per condition plus three embryos without injection in a second experiment ($n = 57$). RNA was extracted from three embryos per condition per time-point (0, 24 and 48 HPI, $n = 54$) plus three embryos per time-point without any injection ($n = 9$, total $n = 63$) as a control for the impact of microinjection.

## *C. auris* strains and growth

We selected five *C. auris* strains to represent each major clade (Data S1.). Culture stocks were kept in $-80\,°C$ in Yeast extract Peptone Dextrose (YPD) broth (Sigma-Aldrich, UK) containing 25% glycerol (ThermoFisher, UK) and plated onto YPD-agar (Sigma-Aldrich) for 48 h before inoculation and overnight growth in 10 mL YPD broth at $30\,°C$ in a shaking incubator at 200 RPM (Multitron, Infors HT, UK). Five mL from each overnight culture was then centrifuged at 5000 g for 2 min before triple washing in 1 mL autoclaved deionised water (DIW, Milli-Q, Merck, USA) to prevent aggregation[28], then re-suspended again in 200 µL DIW. Decontamination of surfaces was conducted with both mopping and cleaning wipes in 70% Ethanol (Sigma-Aldrich) or 10% Chemgene HLD$_4$L (Medimark Scientific, UK) as per manufacturer's instructions. For growth curves, *C. auris* strains were grown overnight in YPD in 4 replicates. Overnight cultures were inoculated into YPD to a final OD$_{600}$ of 0.02 in a 96-well plate. OD$_{600}$ was read over 25 hourly cycles at $37\,°C$ with 300 s shaking at 2.5 mm amplitude on an Infinite 200 PRO plate reader (Tecan, Switzerland).

## Outcome measurement

Individual embryos were kept in 1 mL sterile filtered ASW per embryo in a 48 well flat bottom plate (Corning, USA) sealed with Parafilm 'M' (Bemis Company, USA) at $37\,°C$ with passive humidification using a 500 mL open glass beaker of water. Embryos were monitored for presence of heart rate at 24-hourly intervals up to 7 d by stereo dissection microscope. Initial time of injection was subtracted from each time-point to minimise bias between groups. Colony forming units (CFUs) were calculated by homogenisation of embryos at 0, 24 and 48 HPI by autoclaved plastic micropestle in a 1.5 mL conical tube (Eppendorf, UK) in 100 µL 10,000 U/mL penicillin/streptomycin (P/S) solution (Gibco, UK). Dilutions were made with P/S solution and DIW in 1/10–10,000 and counts were adjusted according to dilution after plating onto YPD agar plates and read after 72 h incubation at $30\,°C$. Statistical testing between CFU numbers was performed with Wilcoxon testing and BH correction.

## Imaging

For direct microscopy, AK were kept in 300 µL ASW containing 0.04% Tricaine anaesthesia (Sigma-Aldrich, UK) in an optical adhesive and parafilm-sealed 96-well plate. These were then monitored for heartbeat by 5 bright field 2X or 4X images at pre-focused z slices taken every 2 h by Acquifer microscopy (Bruker, Germany) over 72 h. All images were processed in ImageJ 1.53a (NIH, USA). Microscopy of YPD-agar-grown colonies was performed without staining in DIW.

## Histology

Sectioning and staining were performed by Microtechnical Services, Exeter, UK. Embryos were fixed in 1 mL 10% neutral-buffered formalin (Sigma-Aldrich, UK) for >24 h, followed by immersion in 4% phenol 10 mL glycerine BP made up to 100 mL with distilled water[126], sectioned at 4 µm without orientation, and stained with Harris Haematoxylin & 1% Alcoholic Eosin on Leica Autostainer. H&E stained histological sections were imaged on an Olympus IX83 microscope coupled with an Olympus DP23 colour camera using an Olympus UPlanXApo x40/0.95 objective controlled by cellSens software v.3.2. Brightfield microscopy was performed with an EVOS M5000 microscope (ThermoFisher, UK).

## RNA extraction and sequencing

Total RNA was extracted using the Monarch® Total RNA Miniprep Kit (NEB, USA). Embryos were flash frozen in liquid nitrogen, re-suspended in DNA/RNA Protection Reagent and disrupted with an autoclaved plastic micropestle. Bead beating was performed in a FastPrep 24 homogeniser (MP Biomedicals) with 0.5 mm zirconia/silica beads for 8 cycles of 35 s at 6 m/s plus 45 s on ice. Lysate was recovered and incubated for 5 min at $55\,°C$ with proteinase K before addition of lysis buffer and RNA purification following manufacturer's instructions. Total RNA samples were quantified using the Qubit 4.0 Fluorometer (Invitrogen, USA), and integrity was checked with the Tapestation 4200 (Agilent, USA). Library preparation and sequencing were performed by the Exeter Sequencing Service. Samples were normalised and prepared using the NEBNext Ultra II Directional RNA Library Prep - Poly(A) mRNA Magnetic Isolation Module following the standard protocol. Libraries were sequenced on a Novaseq 6000 sequencer with 150 bp paired-end reads. Reads were trimmed with fastp[127] v.0.23.1 to remove reads <75 bases and trim based from the 3′ end with q-score <22. Quality control was performed with FastQC (https://github.com/s-andrews/FastQC) v.0.11.9 and MultiQC[128] v.1.6.

## Differential gene expression

We aligned sequences ($\bar{x} = 52.9$m reads per replicate, range 41.1–112.2 m) using the Trinity pipeline[129] v.2.15.1 to the *C. auris* clade I B8441 reference genome (GCA_002759435.2)[29] downloaded from GenBank (mean aligned reads in infected embryos 3.83 m, 0.81–13.1%). We further aligned clade-specific reference genomes for each strain (Data S1.), including clade II (GCA_003013715.2)[4], III (GCA_002775015.1)[14], IV (GCA_008275145.1)[14] and V (GCA_016809505.1)[96,130,131]. We aligned conditions containing *A. dispar* to the de novo GJEY01 adult gill transcriptome assembly[85] downloaded from GenBank, for which we detected expression over 27,481 potential genes. Alignment was performed with Bowtie2[132] v.2.5.1, transcript estimation with RNA Seq by Expectation Maximisation (RSEM)[133] v.1.3.3, differential expression with EdgeR[134] v.3.40.0, Samtools[135] v.1.17 and limma[136] v.3.54.0 using a false discovery rate (FDR) *p*-value cut-off of 0.001 and a log fold change cut-off of $+/-2.0$. Principal components were calculated and plotted using the prcomp function in Rv4.4.0 and the ggfortify v.0.4.17 R package for fragments per kilobase per million, and correlations plots from counts per million as per the default output of the Trinity pipeline. Log fold change was plotted with ComplexHeatmap[137]. In creating volcano plots, we did not display a hypothetical host protein without annotations (HP_DN122662) with a log-fold change of $-11.11$ and a false discovery rate of 30.57 when comparing sham injection *vs* no injection. Additionally, we did not display four *C. auris* transcripts with infinite calculated FDR comparing in-host to in vitro DEG at 24 HPI, including *TNA1_4893* and *NGT1_2864* (up-regulated in-host) and *IFF4_4451* and *ALS4_4112* (down-regulated in vitro).

## Annotation and enrichment

GO terms[138,139] were assigned using DIAMOND[140] v.2.1.8 *via* the DIAMOND2GO pipeline[141] for each of the *C. auris* reference genomes. KEGG terms[142] were assigned using the BlastKOALA online platform using the eukaryote database[143]. Protein FAMily (PFAM) domains[144] were identified using the PFAM-A.hmm database (https://ftp.ebi.ac.uk/pub/databases/

Pfam/current_release/Pfam-A.full.gz) and HMMER and HMM Scan[145] with a cut-off of 1e$^{-5}$. Further domain-specific identification was performed with SignalP 5.0[146], DeepTMHMM 1.0[147] and NetGPI 1.1[148]. Enrichment analysis for DEG subsets was calculated using the Fisher's Exact test, adjusted using the Benjamini–Hochberg method per set of up-regulated or down-regulated features in each comparison as a conservative estimate of significant gene set enrichment, with a false discovery rate cutoff of $p = 0.001$. Annotation tables were checked with additional information from available literature, supplemented by details and web-scraped ortho-logue names from *Candida* Genome Database[82] and Blastp searches[149]. Annotations for the GJEY01 genome were also performed using Dia-mond2GO, BlastKoala and HMMScan as above and checked against *A. dispar* annotations and gene names obtained directly from the GJEY01 transcriptome makers[85]. For plotting, lists of GO terms were identified (and for *A. dispar* enrichment, reduced by removal of redundant terms) by REVIGO[150].

### qRT-PCR

Quantitative PCRs were performed on three replicates of cDNA reverse transcribed from RNA. Equal quantities of RNA from each sample were reverse transcribed using MMLV reverse transcriptase and oligo dT primers (Promega, UK). qPCRs were carried out using PowerTrack SYBR Master mix (Applied Biosystems, UK) with cDNA as template on a QuantStudio™ 7 Pro qPCR platform (Applied Biosystems, UK). Fold change in gene expression level was calculated using the $2^{-\Delta\Delta Ct}$ method with normalisation to the housekeeping genes *B2MG* (beta-2-microglobulin-like, DN135862| c3_g2_i2) in *A. dispar* and *ACT1* (B9J08_000486) in *C. auris*. Statistics were performed using GraphPad Prism v.10.2.1 (calculation of standard deviation). Oligo sequences are as follows (forward, reverse): DN135862| c3_g2_i1 (ATGTCTCTGTGAGGTCACTC, GGGAATTCCTGGACTT-TACG); DN109585|c0_g1_i1 (CCACTAAGAGAAACCACGTC, GGAATTCCTGTCTAGCTCCT); DN112160|c6_g3_i3 (CGGACTG-GAGAGAGAAGATT, ACCTAAGTACCGGGTGTAAG); B9J08_000486 (CGTTGTTCCAATTTACGCTG, TCTCAGCAGTGGTAGAGAAG), B9J08_003921 (GTCTAATCTCCAGACGGTCA, CAGAAGCA-CAAGGGAGTAAC); B9J08_001487 (AATGGTATGTTGAGTTGCCC, ACGTCAAAGGAACCAAGATG).

### Orthologue assignment

To identify accessory genes across *C. auris* clades, we used the Synima pipeline[151] with Orthofinder v2.5.5[152] with additional reference genomes for *C. albicans* (SC5314_GCA_000182965.3) and *C. haemulonii* (B11899_ASM1933202v1). We also incorporated a very recent version of the *C. auris* B8441 assembly (v3) to arrange contigs and illustrate re-arrangements, with broad syntenic concordance; it contains 5 more genes than v2, for which 27 (0.5%) and 16 (0.3%) genes were unique to each genome, respectively. Multiple alignment of single copy orthologues as defined by Orthofinder was performed with Muscle v5.1[153] and a phylogeny was made with FastTree v2.1.11[154] using default settings.

### RNA-seq meta-analysis

NCBI SRA was searched for 'Candida auris' on 16th April 2024 and downloaded as metadata *via* the 'Send to: Run Selector' function. Each project accession number for runs containing RNA was searched against available published literature to identify publications, noting subject of study, clades/ strains, growth media, comparative conditions, temperature and time scales used. Calculation of FPKM was performed and analysed as above using the set of single copy orthologues to demonstrate variance *via* PCA with ggfortify and prcomp packages in R and to calculate an unweighted co-expression Pearson's correlation (cut-off of 0.85) for *SIT1* genes with and without the additional data from the RNA-Seq meta-analysis.

### *SIT1* analysis

The protein transcript for *C. albicans SIT1* was Blastp searched against the draft pangenome database as above. We visualised the structure using

Alphafold v3[155] and ChimeraX v1.8[156]. Clade I-V *C. auris* and *C. haemulonii SIT1* genes were aligned using Muscle as above before using RAXML-NG v1.2.2 with 1000 bootstraps and the LG4X model[157], which was visualised using ggtree v3.12.0. For panfungal *SIT1* analysis, we used the fungidb portal for Blastp using the *C. albicans SIT1* against all available fungal data (with no hits within oomyceta), including 274 reference genomes representing 197 species and a cut-off of 1e$^{-20}$. Annotations for each protein transcript were performed as above to obtain GO terms, KEGG pathways, PFAM domains, GPI-anchors, SignalP domains, and TMRs, while a phylogeny for *SIT1* similar sequences across all fungal references was constructed as above, following multiple alignment with Clustal Omega v1.2.4[158].

### Ethical statement

We have complied with all relevant ethical regulations for animal use. All animal experiments were performed under Project Licence PP4402521 in compliance with University of Exeter Animal Welfare Ethical Review Board regulations and MRC Centre for Medical Mycology Project Review Board, and are reported in line with 'Animal Research: Reporting in vivo Experi-ments' (ARRIVE 2.0) Guidelines[124].

### Reporting summary

Further information on research design is available in the Nature Portfolio Reporting Summary linked to this article.

### Data availability

Raw reads have been deposited *via* NCBI/GEO *via* accession GSE277854. All generated datasets and required to make figures are available *via* Github (https://github.com/hughgifford/Arabian_Killifish_C_auris_2024) and Zenodo[159].

### Code availability

All code required to make figures are available with no restrictions to access *via* Github (https://github.com/hughgifford/Arabian_Killifish_C_auris_2024) and Zenodo[159].

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

## Acknowledgements
We acknowledge funding from the MRC Centre for Medical Mycology at the University of Exeter (MR/N006364/2, MR/V033417/1), MRC Doctoral Training Grants MR/P501955/2 and MR/W502649/1, Wellcome Trust (206412/A/17/Z), Wellcome Trust Career Development Award (215239/Z/19/Z) and Wellcome Trust Fellowship (219551/Z/19/Z), and the National Centre for the Replacement, Refinement and Reduction of Animals in Research (NC3Rs) (NC/X001121/1). T.B. also was supported *via* Elizabeth Ballou and a Sir Henry Dale Fellowship jointly funded by the Wellcome Trust and the Royal Society (211241/Z/18/Z) and the Lister Institute. The views expressed are those of the authors and not necessarily those of the NIHR or the Department of Health and Social Care. We thank staff in the Aquatic Resource Center, including Greg Paull, Paul Tyson, Sam Worthington, Chloe Flint, Jennifer Finlay and Richard Silcox. We also thank the Exeter Sequencing Service facility and support from Wellcome Trust Institutional Strategic Support Fund (WT097835MF), Wellcome Trust Multi User Equipment Awards (WT101650MA and 218247/Z/19/Z), Medical Research Council Clinical Infrastructure Funding (MR/M008924/1) and BBSRC LOLA award (BB/K003240/1). We also thank the University of Exeter High-Performance Computing (HPC) facility funded by the UK MRC Clinical Research Infrastructure Initiative (award number MR/M008924/1). We thank Tanmoy Chakraborty, Dhara Malavia and Neil Gow for providing *C. auris* strains, Teigan Veale, Sumita Roy, Jonathan Ball and Darren Thomson for assistance in microscopy and Chloe Pelletier for advice on aggregation. We also thank Microtechnical Services, Exeter, UK, for assistance with histological sectioning.

## Author contributions
H.G.: designed and conducted experiments, performed analysis, drafted and wrote the paper. N.H., J.G., Q.M. and T.B.: optimised, designed or performed experiments. R.F., T.K., T.B., M.R., J.R., D.W. and A.B.: designed experiments, obtained funding. All authors edited the manuscript.

## Competing interests
The authors declare no competing interests.
