## [Transparent Peer Review file · Communications Biology]

Xenosiderophore transporter gene expression and clade-specific filamentation in *Candida auris* killifish infection

Corresponding Author: Dr Rhys Farrer

This manuscript has been previously reviewed at another journal. This document only contains information relating to versions considered at *Communications Biology*.

Version 2:

Reviewer comments:

Reviewer #2

(Remarks to the Author)

I consider my previous comments on this manuscript to be addressed in regards to this revised version.

Reviewer #3

(Remarks to the Author)

Congratulations to the authors for this comprehensive revision of the manuscript. Verifying the gene expression ensured the RNAseq finding of key genes. However, certain aspects of the current study remain unaddressed in this paper. I understand gene KO, drug inhibitor studies, verifying identified key genes in a more clinically relevant in vivo model, or even ex vivo, and other suggested experiments are not feasible due to the obstacles described by the authors. At least a lack of these important experiments to confirm the role of the reported genes should be described as limitations of the study in the discussion section.

Reviewers' Reply

Reviewer #1 (Remarks to the Author):

In this paper, Gifford and colleagues seek to understand the transcriptional response by both fungus and pathogen in a novel Arabian Killfish model of infection. They first demonstrate that they can achieve virulent infection in this model, and observe slight differences between strains in terms of mortality kinetics. They then use dual-RNAseq of both the pathogen and the fish to understand pathways that are enriched upon infection. They focus on expansion of a family of siderophore transporters and implicate this process in disease.

Thank you very much for this summary of our paper.

There are some major limitations of this study that should be addressed.

1) What is the correspondence between the killfish model and mammalian systems? For example, although the authors point to the presence of IL-17 signaling in fish as a reason why they are a good model for infection, none of the IL-17 signatures are observed in the Ak egg model.

We have clarified that IL-17 activation is a feature of adaptive immunity which we would only expect to observe in adult fish (line 115). Thus, the lack of IL-17 responses is a limitation of embryo models in general, since larvae typically employ innate immune defences (lines 113-115). We have clarified this and outline in our introduction that, alongside the model's potential, these limitations offer "a balance between considerations such as suitability for experimental manipulation and applicability to human fungal infection at mammalian temperatures" (lines 119-120).

Potentially a systemic infection would be more accurate, but overall, more information is needed before it is clear that this is a useful system. This should include a detailed comparison between the gene expression in Ak egg and the different mammalian and cell models of infection to identify what is specific to fish and what is a general response. Without this information, the utility of the model is unclear.

Our paper seeks to demonstrate the utility of the AK embryo system for one specific purpose: describing an in-host living tissue transcriptome of *C. auris* infection, the most urgently lacking gap in the literature. These are excellent ideas for further work to develop and understanding differences among model/host organisms. We now outline these limitations in our introduction (lines 78-90).

It is unclear from the above comment whether the reviewer is suggesting further expression comparisons amongst different hosts or different pathogens, or perhaps both. To determine how the pathogen responds to different hosts and environments across different experimental setups, we compared global public transcriptome data for *C. auris* (supplementary table 4, supplementary figures 4E-

4F). We used our RNA-seq pipeline to analyse 35 separate datasets and combined these with insights from systematic review of each individual paper to outline the range of different settings and *C. auris* clades used in each experiment (supplementary table 4). We found that across these settings, through principal components analysis (supplementary figures 4E) and a heatmap of per-gene variance for each group of studies (supplementary figures 4F) that the Killifish dataset is not an anomalous outlier, indicating commonalities among transcriptional response in *C. auris*.

2) For the mortality data, it is not clear how certain strains can be significantly different from each other given overlapping mortality curves.

Our Kaplan-Meier curves show red (clade I) and green (clade IV) curves fall to a lower survival before other clades, especially between 48 and 120 hours, as can be seen on direct inspection of the graphs. The Log-Rank test determines significance between survival curves (<https://pmc.ncbi.nlm.nih.gov/articles/PMC403858/>), which remains significant even after testing for multiple correction (Benjamini-Hochberg). We have used standard methodology including statistical tests, the results of which is in line with expected mortality differences, as detected by other studies which we have cited (lines 73-76). We planned for and anticipated a significant difference between clades, and we used the widely used power calculation methodology to arrive at the number of embryos required (lines 459-466).

3) In dual RNAseq, a major limitation is lack of pathogen signal, and in the methods section, the authors describe an average read depth of 3.8 m from infected embryos. Is this sufficient coverage, especially to compare to the in vitro cultures?

A read depth of 3.8 million paired reads (760 Mb of sequence) across the transcriptome is more than sufficient and far beyond the depth of sequencing achieved in many other *in vivo* studies. With a transcriptome encompassing only ~8.1 Mb, our paired end sequencing amounts to ~94X deep.

The tables for RNAseq analysis (Supplemental Tables 2 and 3) should have not just whether the gene was significant, but also the FPKM in each strain and the log2 fold change for each strain. If the coverage is too low, it will confound the differential analysis.

We have now included this data in supplementary tables 2 and 3. Further data is available via the data repository for the paper (https://github.com/hughgifford/Arabian_Killifish_C_auris_2024) in the data folder (https://github.com/hughgifford/Arabian_Killifish_C_auris_2024/tree/main/data).

4) Whether the DEGs in these tables were decided using the Clade I reference or the appropriate strain references is not clear. Each strain transcriptome should be compared to its appropriate reference. Therefore, lines 259-302 should probably be removed and only the re-analyzed data included.

We have re-iterated which reference genome is used. The use of the B8441 reference genome is a typical benchmark across genomic studies of *C. auris*. This is especially important for an emerging pathogen, and a useful method to compare individual strains' expression. This approach can capture "core" gene expression but is limited in accessory gene expression. Therefore, we took two approaches:

- 1) to base our enrichment approaches on clade-specific reference genome transcriptome calling.
- 2) we use FPKM between orthologous groups to confirm or refute hypothetically significant differences

The inclusion of both approaches enables easy communication across the discipline and freedom from confusion. Additionally, given the publication of an updated reference genome for *C. auris*, we have included a new table (Table 1) into the manuscript, to show the proposed name for each gene highlighted in our study, the chromosomal locations, and the code names from the most widely used reference (B8441) and its updated code (B8441 version 3). We see these additions as being important for the maximal impact of our findings across the field by retaining familiar and searchable/findable gene codes, since literature reviews in the future may rely on either.

5) There is a lot of speculation about the function of the XTC and SIT transporters given a potential signature for expansion. The authors should reference PMC10319987 when discussing gene family expansion in *C. auris*.

Thank you for highlighting this paper. We agree that linking to the field of gene family expansion research in *C. auris* is important for the impact of this paper. We have prioritised an inclusion of an earlier key reference in the *C. auris* literature on gene family expansions, namely this paper (<https://pubmed.ncbi.nlm.nih.gov/30559369/>), which mentions specifically the expanded gene family of *SIT* orthologs (Fig. 4b) that we explore in this paper, as well as addressing some of the adhesin families (HYR/IFF) that were explored by Smoak and colleagues (PMC10319987). For clarity, we have not included the suggested reference for these reasons. However, if the journal would be happy with the increased length of the article as a result, we would be happy to include this article as an example of another expanded gene family within the discussion.

Additionally, do these genes also show an upregulation in other models of infection? If this expression pattern is only in the fish, this may not be useful for understanding mechanisms of pathogenesis. To really demonstrate that a particular gene process is important for infection, the authors should take advantage of some of the many ways to genetically manipulate *C. auris* and show that the *SIT* or *XTC* genes are important.

Yes, these genes do see an upregulation in other models of infection, as mentioned above. We agree that genetic manipulation of *SIT/XTC* genes is warranted, which we detail below.

6) For the co-expression, the authors should include the underlying data and the benchmarks used to test for co-expression. More information is needed to evaluate this. For reference, PMID 33472984 and PMID 37645941 demonstrate

the benchmarking needed to demonstrate sufficient data for a robust expression analysis in fungi.

Thank you for highlighting these two excellent studies. The inference of a co-expression network requires detailed analysis and there are multiple approaches to these well-established methodologies. However, in our study, we examined the co-expression for *SIT1* genes only and did not undergo weighted or modular network analysis. Our code is available as detailed in the article and as checked by Reviewer #3.

We would be glad to use these techniques in follow-up work, but as you have indicated, such an analysis would require at least a whole manuscript to demonstrate a robust network analysis across the whole transcriptome. In this case, we sought to understand highly correlated expressed genes related to the *SIT1* expanded gene family, rather than complete the standalone work of building a co-expression network with hubs and modules.

As a result, we have moved these analyses to supplementary data and removed any hint of a suggestion that they are “network” analyses, instead referring to these as a simple assessment of potentially directly co-expressed genes. The rationale for comparing different studies using different tissues, timepoints, multiplicities of infection and media, needs to be carefully considered.

Minor points:

Figure 2: How do D and E relate to A? The supplemental data table does not include this information, so any further analysis of a specific gene is limited.

In our revised manuscript, this is now figure 3. The legend explains that A shows the total numbers of DEGs, while D-E show the log-fold change and significance level for these differences. We have included the log-fold change data in the supplementary table 2, which included detailed information about each gene, and in which clades differential expression was significant.

Figure 3: It is difficult to use the data as it is currently presented in this figure. The number of differentially regulated genes is presumably less important than the specific identity of these targets. Additionally, does 3F use data averaged across strains? If so, what happens to genes that are up in one strain but down in another?

In our revised manuscript, this is now figures 4 and 5. We used a simple bar chart (Fig. 3A-D, now figure 4A-B and figure 5) to give an overview of experimental conditions and develop the rationale for focusing on specific comparisons. Fig. 3F (now 4C) does use averaged data, since this straightforwardly highlights genes that share common expression patterns across the species. An example of a gene which was “up in one strain but down in another” is *SCF1*, which can be seen in the centre of the volcano plot (mean log-fold change across clades is thus approx. zero) on the right-hand panel of Figure 4C. The specific datapoints and genes for each experiment are available in both supplementary tables as mentioned above and in the Github data repository. To concisely communicate the gene expression patterns

across the species, the use of mean log-fold change is a clear data visualisation that separates out genes that are “up in one strain but down in another” (tends towards a mean of zero) and up in all (e.g. *SIT1_1487*, tending towards a positive mean).

Reviewer #2 (Remarks to the Author):

Here Gifford and colleagues examine strains representing different clades of *Candida auris* in the killifish larval model. Their examination includes basic pathogenicity measures in killifish followed by transcriptomic analysis of host and pathogen. While demonstrably new model and with some interesting findings the manuscript as it stands is only a small step forward in our understanding of *C. auris* as a pathogen.

Thank you very much for kindly saying that this is interesting and a step forward, we are very grateful. However, it is important to clarify that the novelty of this study is not that we are justifying or presenting merely a new model *per se*, but rather that we are using this new model to explore *C. auris* in-host tissue infection gene expression for the first time as a sorely lacking gap in the science. In our study, we report RNA-seq from a pathogen during live host tissue infection that is the major step for this work, and less obtainable from several other models. For further work regarding the AK model, please see Atyaf Hamied’s PhD thesis from 2018 (<https://ore.exeter.ac.uk/repository/handle/10871/33664?show=full>) or a pre-print looking at *C. albicans* infection (<https://www.biorxiv.org/content/10.1101/2024.10.08.617174v1.full>). Neither previous work examines the pathogen transcriptome, nor the use of *C. auris* infection in any way at all. There is no overlap in the paper under review or any of these (or any other) studies in terms of experiments, data, text or work.

1. There is mention of power calculation in the methods but there is no justification of the statistical analysis used. For example, the groups sizes for infection experiments were based on an independent T test but a mathematically distinct Kaplan-Meier method is used for significance testing. Furthermore, what was the multiple comparison adjustment?

We used two separate tests (a power calculation before planning animal/human experiments, followed by a survival analysis after animal/human experiments), which is ethically required. We have ensured our justifications are now more prominently shown. Specifically, we used independent t-tests for power calculation to decide the number of embryos needing to be injected *before* the study, according to the guidelines from NC3R, who part-funded this research (<https://eda.nc3rs.org.uk/experimental-design-group>), and ARRIVE guidelines (referenced in methods). Then after the study, survival data analysis using time-series data required a statistical method that considers censoring *via* comparison of probabilities of death at any one time, for which Kaplan-Meier analysis is a foundational method. Multiple comparison adjustment is clearly stated as Benjamini-Hochberg.

2. Even making the assumption that the Kaplan-Meier analysis is solid what do these differences survival mean for *C. auris* pathogenicity? How do they relate to clinical data?

We reference the only study to have systematically analysed clinical data in original lines 78-80, discussed above. Clinical data is complicated by the tendency of clades I, IV and III to cause bloodstream infection and others to have mostly been demonstrated only in non-fatal otomycosis (lines 71-73). However, invasive candidiasis is complicated by a range of factors – the tendency of individual strains to cause outbreaks in hospital settings, profiles of resistance, adhesion to indwelling blood and urinary catheters, etc. As a result, our study pinpoints the in-host question of what characterises gene expression when *C. auris* is surviving and replicating in a living host tissue.

Was a dose response curve performed. This is a critical point as all the following analysis are based on this model.

We tested doses between 50, 500, 1000 and 2000 yeasts in pilot experiments for clade I infection and ascertained a suitable dose for a transcriptome data. In line with the NC3Rs principles, we sought to refine and reduce animals and so used the minimum required experimentation to achieve an *in vivo* transcription experiment to identify common gene expression signatures during infection. The priority of this study was to achieve gene expression analysis, which we have now demonstrated, and can use as a benchmark for future studies in human hosts as we seek to understand the pathogen transcriptome in host tissue.

3. It is of potential interest to find no difference in cfu but a difference in virulence. There are many methods for directly relating these in the animal infection literature. As in point 2, given the aim of this study to relate transcriptional differences to pathogenicity this is a major failing. Figure 1c goes toward this but it is again very cursory. Why no higher resolution time lapse imaging of infection progression? This is reason for using a translucent model organism and the advantage of studying fungal pathogens is they are large enough to see without a stain or fluorescent marker.

There was only a difference in CFU between clades III and V, which we hypothesise could be related to filamentous aggregation (original lines 500-501). Time-lapse imaging for survival curves often uses shorter timescales (hours-to-days, e.g. <https://www.biorxiv.org/content/10.1101/2024.10.08.617174v1>) and we found that a seven-day experiment was required, which makes fine-resolution timelapse imaging less practical. We agree that embryo translucency holds excellent promise for the AK model, alongside knockouts and fluorescent cell lines. However, as with the zebrafish model, these model developments will require years of work to fulfil such potential. Work is ongoing to obtain fluorescent cell lines for phagocytes and other cell types in the Arabian killifish. For interest, the link to our grant-funded work in this area is here: <https://gtr.ukri.org/projects?ref=NC%2FX001121%2F1> :- the goals of this research include the generation of transgenic cell lines for 4d imaging during fungal infection (e.g. with fluorescent phagocytes). The goal completion date for this grant activity is March 2026. The findings of this paper, coupled with the potential of 4d modelling in the embryo, could be combined in further studies, increasing the impact of this paper.

4. The xenosiderophore findings are of potential importance but little can be concluded from the data presented. Much further work is needed: generation of knock out mutants, reconstitution between different clades etc.

Thank you for this suggestion. As is the case with the *ALS* gene family in *C. albicans*, many years and many publications have been required to understand the multiple, overlapping and complex roles of an expanded gene family. For example, the memorable work of Lois Hoyer and others from 2008 (where each member of the gene family is given a cartoon characterisation, PMC2742883) cites over 80 papers used to build up this picture for each gene family member (and work is ongoing). Our study provides a foundation to conduct a broad programme of investigation as a next step building on what we have established. Validating the roles of these novel gene families are beyond the scope of a single publication. We have now made greater emphasises for the need for further experimental validation in our discussion.

5. It concerns me that there wasn't much to be told about the host response. As the authors state: 'It is notable that the yolk-sac model was not associated with histological or transcriptional evidence of host neutrophil recruitment and activation, a prominent feature of human blood infection modelling.' Is this due to a deficiency in the model or methods or something else?

We have rephrased our work to clarify that analysis of the host response was a secondary consideration. Please refer to our response to R1, Q1. Living hosts contain many varied tissues, which *C. auris* can infect. Our findings of pathogen *XTC* upregulation across the species are in line with the upregulation of these genes in the whole blood infection model. This hints at the clinical relevance of this model for understanding pathogen gene expression at mammalian body temperature during infection. We do not confuse the nature of the yolk-sac as a body of tissue in a developing embryo; there, innate immune effectors such as antimicrobial peptides are an expected response to infection, rather than the gross and overwhelming systemic haematogenous neutrophilic activation and migration found in human sepsis. We note in our paper the discussions around *C. auris* being immunoevasive; additionally, several zebrafish injection studies have identified *C. auris* neutrophilic activation *via* hindbrain injection rather than yolk sac injection. We recognise these as features of a developing embryonic fish larval model. Comparison with the many specific microenvironmental niches in the clinical setting, with varying degrees of immune privilege, is awaited.

6. The legends are lacking in sufficient detail of the experiments, data and statistical analysis.

We submitted our article in line with Nature submission guidelines (<https://www.nature.com/nature/for-authors/formatting-guide>): "Each figure legend should begin with a brief title for the whole figure and continue with a short description of each panel and the symbols used. If the paper contains a Methods section, legends should not contain any details of methods." However, we have now largely re-written most of our manuscript, including revising the figure and table legends. We have updated legends to include as much information as possible, as requested. Please clarify whether legends should include further details of individual

experiments – each should now contain all the required detail.

Reviewer #3 (Remarks to the Author):

This manuscript by Gifford et al. describes the pathogenic mechanisms of *Candida auris*, a significant fungal pathogen identified by the WHO as a critical priority due to its rising drug resistance and high mortality rate (~45%). The study introduces a novel fish embryo yolk-sac microinjection model using the Arabian killifish to mimic human body temperature, allowing authors to analyze the transcriptional responses of *C. auris* during in vivo infection. The research employs dual host-pathogen RNA sequencing at 24 and 48 hours post-injection, revealing key host gene expressions related to heat shock, complement activation, and nutritional immunity. Notably, the study identifies a new family of xenosiderophore transporters (XTC) and a sub-clade of haem transport-related (HTR) genes that are up-regulated during infection. The findings suggest that these genes may be crucial for the virulence of *C. auris*, highlighting their potential as targets for future therapeutic strategies.

The strength of this study includes the comprehensive analysis of both host and fungal gene expression analysis, which is crucial in understanding *Candida auris* virulence and its effect on the host. Also, the authors established an innovative Killifish embryo yolk-sac infection model to carry out these studies. This model can be employed in antifungal research and could be an essential resource for the anti-fungal research community.

Thank you for highlighting the strengths of this study.

However, the Killifish embryo-sac infection model has limited clinical relevance for studying *C. auris* pathogenic and immune responses.

We agree that there is limited clinical relevance. We have been more careful with our language regarding “clinical relevance” and ensured limitations are clear.

Yet, these studies are crucial to understanding the *C. auris* virulence mechanisms and host responses and will be quite resourceful to the research community.

Thank you. We agree that this study is a significant step forward and a resource for the wider community.

Major Critiques:

The Killifish embryo-sac model is innovative, providing insights into *C. auris* gene expression in a controlled environment that simulates human conditions. However, this model is not clinically relevant and lacks an appropriate host tissue environment and immune system similar to humans. Furthermore, the lack of information on antifungal immune responses (specifically innate immune cells and inflammatory/regulatory cytokine responses) in this model and their similarities/ dissimilarities with humans or even with more clinically relevant models such as mice should be highlighted.

Thank you for this comment, which is like R1, Q1-2, which we have replied to above. We would not expect an adult immune response in a developing larva, and the urgent question is not what the host immune response features (since several studies explore this) but the gap of pathogen transcriptomics *in vivo*.

We have rephrased our explanation as “Given that *XTC* gene up-regulation by *C. auris* has also been detected during *ex vivo* whole blood infection (*XTC2A-B*, *B9J08_1547-8*), this suggests that our findings may be clinically relevant”. For further instances, see several specific changes highlighted below.

In terms of highlighting the similarities between models, we have ensured our introduction is a streamlined navigation through the gaps in the literature, the problems faced in methodology, and the reason for the use of a limited model.

Thangamani et al. (PLoS Pathogens 2024) recently published a host transcriptomic analysis at the skin interface utilizing a mouse intradermal infection model. The authors should discuss their findings related to host responses in the context of these more clinically relevant models and highlight significant differences and limitations of their model.

Thank you for highlighting this study, which details the use of a skin infection model to understand host responses in adult humans is excellent. However, host gene expression during *C. auris* infection was only ever a secondary aim for this paper, as discussed above. We could compare other studies examining the host response, but the expected differences between adult human tissues and a developing fish embryo response is likely nuanced (including with regards to developmental stages, which tissues). Their study does not address pathogen gene expression, which is the major gap in understanding *C. auris* pathogenicity and infection. Notably, this study does not compare between multiple host models or cell lines, nor does it examine pathogen gene expression, nor does it use more than one *C. auris* clade. We believe that a reason is likely to be low numbers of fungal cells and limited. Hence, the AK yolk sac model provides an opportunity to study gene expression with more robust sequencing support. However, while this may be the case in a bulk sequencing experiment, using the AK model in future using 4d methods combined with, for example, single cell sequencing, it may be possible to ascertain specific host cell type responses, which would enable the sort of granular comparison required to understand gene expression programme differences.

Figure 1A shows Clade III as the least virulent but with a significantly higher fungal burden. The author does not discuss the potential explanation and implications of this significantly higher fungal burden for the host gene expression profile.

We address this consideration for R2, Q3. This should be rephrased as an apparently lower fungal burden in clade V.

Further, it is essential to confirm the role of the central virulence genes *XTR* and *HTR* identified in this study in a clinically relevant model. If this is logistically not possible at this stage, the functional role of *XTR* and *HTR* in *C. auris* could be further analyzed by comparing iron supplementation and

chelators on *C. auris* pathogenesis in the Killfish model. The fungal and host gene over-expression should be confirmed by additional methods, such as quantitative PCR, to validate the outcome of transcriptomics analysis. Next, the top identified upregulated host (e.g., heme Oxygenase) and *C. auris* genes (XTR and STR genes) in the Killfish infection model are only suggestive of playing a role in the virulence/ pathogenic mechanisms through correlation and need further confirmation. The functional role of these genes in virulence /pathogenesis should be confirmed by conducting additional experiments in which infected Killfish can be treated with iron supplementation vs iron chelating /siderophore inhibitors and subsequently determining the impact of these treatments on survivals and fungal burden. Further, in vitro iron acquisition assay in the presence vs absence of siderophore inhibitors could confirm the role of XTR and STR in *C. auris*. Similarly, the role of significantly upregulated host genes (e.g., HMOX) in *C. auris* infection could be confirmed using gene knockout or heme oxygenase inhibitors (e.g., Imidazole–dioxolanes) during *C. auris* infection and potential impact on Killfish survival/ fungal burden.

Thank you for these excellent suggestions for follow-up experimental work. We have now completed the requested PCR experiments, including for two host genes (*HMOX*), DN109585 and DN112160 during infection (lines 187-190) and for two pathogen genes (*SIT*), *XTC3* (B9J08_003921) and *XTC7* (B9J08_001487) during infection (lines 207-211). These results confirm our findings for a specific interaction (clade IV infection). We additionally identified necessary housekeeping genes, which we report in our methodology. Furthermore, qPCR is less sensitive and detailed as the results we obtained from RNAseq and therefore has provided only further validation (aside from multiple technical and biological repeats, experiments in different clades, comparison to the *ex vivo* model, etc.).

This list of suggestions by the reviewer covers a wide range of experimental methods, any one of which would be worthy of new bodies of work. We would need to address a number of specific features – the use of iron chelators/siderophores as a drug on the Arabian killifish embryo (subject to the necessary ethical approvals required to undertake such work); the pharmacokinetics and pharmacodynamics of iron chelators in the embryonic yolk sac; the relationship of these drugs spatiotemporally in the yolk sac compare to the yeast; the effect on CFU, growth rate, and gene expression; a range of iron chelators as we now mention in lines 416-419, the specific composition of the embryonic yolk sac e.g. by mass spectrometry; how the yolk sac then compares to other tissues inc. human tissues in terms of iron availability; as well as other considerations for host responses in 4d with fluorescent cell lines as mentioned above. In terms of knockouts, further funding and ethical approval (see above work detailing the NC3Rs grant for the Arabian killifish transgenic lines) would be required (in the UK, Home Office licences are required for transgenic line creation in fish). All the above mean that work to undertake drug injection or new transgenic line creation in Arabian killifish is not logistically possible at this stage. Of course, we agree these are exciting lines of enquiry.

In terms of knockouts for *C. auris*, we mention above the care required to undertake knockouts for expanded gene families, where redundancy and interplay require a multi-pronged approach in different settings, as we allude to by citing the

work of Lois Hoyer and others in relation to deciphering the roles of the *ALS* gene family in *C. albicans*. As a result, knock-out experiments for *C. auris* *XTC/HTR* genes would be a major undertaking, and would distract from the major thrust of this paper, which sheds light on *C. auris* gene expression in a living host tissue for the first time.

Finally, the manuscript could benefit from a clearer discussion of the limitations and implications of this study's findings. While it mentions the potential of XTC and HTR genes as therapeutic targets, it does not elaborate on how these findings could translate into clinical applications or influence treatment strategies.

We have re-phrased accordingly. We have elaborated briefly on *XTC* and *HTR* genes in the light of further emerging evidence. We need to carefully consider the need to tone down the clinical importance of our findings. We exercise restraint in discussing newly reported findings (lines 416-419) showing synergy of echinocandin therapy with iron chelation. These exciting pre-clinical findings bridge our work with clinical applicability, but we will not oversell the clinical applicability of our findings. Instead, we recognise that iron chelation in a clinical setting is complicated by the iron status in the critically ill patient, the need to balance oxygen delivery and haemoglobin levels with pathogen iron demands, the emerging evidence around optimal haemoglobin levels in critical illness, the role of iron deficiency anaemia in acute and chronic illness, and the complicated nature of manipulating host iron availability therapeutically. Instead, we follow-up our focus on the evolutionary dynamic of these *C. auris* gene family expansions in relation to the fungal kingdom, which is particularly important in relation to the question of the emergence of novel critical priority fungal pathogens with global public health impact.

Minors comments:

The authors used many software tools, and reading through the text about these tools is overwhelming. These must be tabulated with multiple columns such as tool name, purpose/uses, brief description, source, and references. Similarly, the data analysis pipeline should be depicted in a graphical format to help the readers understand the underlying data QC, analytical qualification, and statistical methods.

The number of items in our study is now near the journal limit, and we aimed to include a representation of key analyses. We have now included a new Figure 2 to include a graphical format of the pipeline as requested. Should a table be necessary in addition to this, please would you clarify if this should sit in the main text or in the supplementary files. It is most common for bioinformatic papers to report version only in the methods, to our knowledge.

There are typographical errors, e.g., repeated words in line 591 and lines 705-706.

We could not see unnecessary repetition in original lines 705-6.

Reviewer #3 (Remarks on code availability):

The codes and source file look ok and appropriate information in the ReadMe file is included to replicate the analysis. No other issues were identified.

Thank-you for looking at our code. This has been updated recurrently to ensure continuity with the text.